# Convergence-Rate-Matching Discretization of Accelerated Optimization Flows Through Opportunistic State-Triggered Control

**Miguel Vaquero**
Mechanical and Aerospace Engineering
UC San Diego
San Diego, CA 9500
mivaquerovallina@ucsd.edu

**Jorge Cortés**
Mechanical and Aerospace Engineering
UC San Diego
San Diego, CA 9500
cortes@ucsd.edu

## Abstract

A recent body of exciting work seeks to shed light on the behavior of accelerated methods in optimization via high-resolution differential equations. These differential equations are continuous counterparts of the discrete-time optimization algorithms, and their convergence properties can be characterized using the powerful tools provided by classical Lyapunov stability analysis. An outstanding question of pivotal importance is how to discretize these continuous flows while maintaining their convergence rates. This paper provides a novel approach through the idea of opportunistic state-triggered control. We take advantage of the Lyapunov functions employed to characterize the rate of convergence of high-resolution differential equations to design variable-stepsize forward-Euler discretizations that preserve the Lyapunov decay of the original dynamics. The philosophy of our approach is not limited to forward-Euler discretizations and may be combined with other integration schemes.

## 1   Introduction

This paper builds on the current research activity that seeks to characterize the convergence properties of dynamical systems that are continuous-time versions of accelerated algorithms in optimization. This body of work sits at the intersection of various disciplines, most notably nonlinear systems and optimization, and has brought to the understanding of acceleration properties a wealth of powerful techniques from Lyapunov stability analysis, calculus of variations, and geometric methods. This paper takes another step in this direction by further advancing the synergy between stability analysis and the study of optimization algorithms. Here, we propose to employ an opportunistic state-triggered approach to discretize continuous flows in a way that respects the Lyapunov function decay that explains their accelerated convergence rates.

**Summary of Results**

The contribution of this paper is the design of a variable-stepsize forward-Euler discretization that preserves the Lyapunov decay of high-resolution differential equations. A main novelty of our technical approach is to employ, in the context of the discretization of state-of-the-art optimization flows, ideas from opportunistic state-triggered control to develop real-time implementations of closed-loop dynamical systems. We build on the Lyapunov functions employed to characterize the rate of convergence of high-resolution differential equations to identify triggers that help us determine the stepsize of the discretization as a function of the current iterate. By design, these triggers ensure that the discretization retains the decay rate of the Lyapunov function. Since the

evaluation of the Lyapunov function relies on knowledge of the problem optimizer, we rely on well-known bounds available for strongly-convex functions to synthesize triggers that do not require such knowledge. Various simulations show the superior performance of the proposed method in comparison with recently proposed constant-stepsize discretizations. The flexibility of the proposed framework provides a promising path towards the understanding of the acceleration phenomenon and the design of new adaptive optimization algorithms.

**Related Work**

**State-Triggered Control.** The basic idea of opportunistic state-triggered control, see [13, 20] and references therein, is to abandon the paradigm of continuous or periodic sampling/control in exchange for deliberate, opportunistic aperiodic sampling/control to improve efficiency in the use of resources while maintaining stability. Opportunistic state-triggered control can be roughly divided into event-triggered and self-triggered designs. In event-triggered control, one continuously monitors certain conditions whose violation triggers certain desirable action whereas in self-triggered control the aim is to predict, with the information available at the last triggering time, when the next triggering condition will take place. Beyond stability, many triggered designs also pay attention to guaranteeing a desired performance by, for instance, making sure that the system enjoys a certain convergence rate. This is accomplished through careful analysis of the evolution of a Lyapunov function. Works [6, 12] based on the *derivative-based* approach ensure the closed-loop system's stability by monitoring the time derivative of the Lyapunov function. Other works [7, 26] resort to *Lyapunov sampling*-type conditions, where triggers are stated in terms of the Lyapunov function by monitoring its decay. Recent work [6, 21] combines the strengths of both types of design. Our technical treatment here follows the derivative-based approach, albeit we believe that other types of design could also be combined with our results here.

**Accelerated Methods in Optimization.** Steepest gradient descent is a keystone in first-order optimization methods, but can be very slow. The work [22] introduced the so called *heavy-ball method*, which aims to speed up the convergence of the gradient descent algorithm by including a momentum term. Later on, [18] designed an algorithm similar in form, the so-called *Nesterov's accelerated gradient*, and using the technique known as *estimating sequences*, showed that the method achieves black-box oracle bounds, i.e., it is optimal on the class of smooth convex or strongly convex functions. Ever since its appearance, acceleration has remained mysterious, to a great extent due to the elegant but unintuitive algebraic arguments used by Nesterov in his derivations. To clarify the ideas underlying acceleration methods, the literature has explored different viewpoints. Some work [1] relies on coupling different dynamics, where at any step mirror descent and gradient descent are interpolated. Other approaches are based on dissipativity theory, [14], integral quadratic constraints, [16], and even geometric arguments [5]. The most relevant line of research for our purposes is the one initiated in [25], which introduces a second-order differential equation which is the continuous limit of Nesterov's accelerated gradient method. This ODE exhibits approximate equivalence to Nesterov's scheme and thus can serve as a tool for its analysis. Especially salient is the fact that the analysis (both stability and rate of convergence) of the mentioned ODE is carried out using a Lyapunov function. This work has spurred a lot of activity aimed at uncovering the rationale behind the phenomenon of acceleration resorting to continuous dynamics, including the variational viewpoint introduced in [27], the connections between Lyapunov theory and estimating sequences in [28] and the Hamiltonian perspective exploited in [8, 17]. The work [15] employs a hybrid systems approach to design a continuous-time dynamics with a feedback regulator of the viscosity of the heavy-ball ODE, guaranteeing arbitrarily fast exponential convergence. Recently, *high-resolution ODEs* were introduced in [23] as more accurate surrogates for the heavy-ball and Nesterov's algorithms. The work [3] introduces similar dynamics under the name *inertial systems with Hessian-driven damping*. A number of works have also explored the discretization of accelerated continuous models and their stability. The work [27] shows that the forward Euler method can be inefficient and even become unstable after a few iterations. Some experimentation using symplectic integrators, without theoretical guarantees, is given in [4]. The work [29] shows that high-order Runge-Kutta integrators can also be used to retain acceleration when discretizing Nesterov's methods for convex functions. The work [24] analyzes in detail the properties of explicit, implicit, and symplectic integrators when applied to the high-resolution dynamics corresponding to the heavy-ball and Nesterov's schemes. The methods proposed in this paper can be understood as variable-stepsize discretizations, which are a popular class of methods in numerical analysis. Some examples of their success include line-search methods

in optimization [19], the Runge–Kutta–Fehlberg algorithm [11], and adaptive-structure-preserving integrators [10].

## 2 Preliminaries

### 2.1 Notation and Assumptions

We denote by $\mathbb{R}$, $\mathbb{R}_{>0}$, and $\mathbb{N}$ the sets of real, positive real, and natural numbers, resp. All vectors are considered column vectors and we denote their scalar product by $\langle \cdot, \cdot \rangle$. We use $\|\cdot\|$ to denote the 2-norm in Euclidean space. Given $\mu \in \mathbb{R}_{>0}$, a function $f : \mathbb{R}^n \to \mathbb{R}$ is convex if $f(kx + (1-k)y) \leq kf(x) + (1-k)f(y)$ for $x, y \in \mathbb{R}^n$ and $k \in [0, 1]$. A continuously differentiable function $f$ is $\mu$-strongly convex if $f(y) - f(x) \geq \langle \nabla f(x), y - x \rangle + \frac{\mu}{2} \|x - y\|^2$ for $x, y \in \mathbb{R}^n$. Given $L \in \mathbb{R}_{>0}$ and a function $f : X \to Y$ between two normed spaces $(X, \|\cdot\|_X)$ and $(Y, \|\cdot\|_Y)$, $f$ is $L$-Lipschitz if $\|f(x) - f(x')\|_Y \leq L \|x - x'\|_X$ for $x, x' \in X$. We endow the space of $\mathbb{R}^{n \times m}$ matrices with the induced matrix norm, namely $\|A\| = \max_{\|x\|=1} \|Ax\|$. We denote by $\mathcal{S}^1_{\mu,L}(\mathbb{R}^n)$ the set of continuously differentiable functions on $\mathbb{R}^n$, $\mu$-strongly convex that have $L$-Lipschitz continuous gradient. The function class $\mathcal{S}^2_{\mu,L}(\mathbb{R}^n)$ is the subclass of $\mathcal{S}^1_{\mu,L}(\mathbb{R}^n)$ of twice differentiable functions with Lipschitz Hessian. A function $f : \mathbb{R}^n \to \mathbb{R}$ is positive definite relative to $x_*$ if $f(x_*) = 0$ and $f(x) > 0$ for $x \in \mathbb{R}^n \setminus \{x_*\}$.

### 2.2 Opportunistic State-Triggered Control

Here we provide a basic account of how real-time implementations of continuous-time controlled dynamical systems can be developed using opportunistic state-triggered control. We refer to [6, 13] for more complete expositions. We build on these ideas later to develop discretizations of high-resolution differential equations. Consider the controlled dynamical system on $\mathbb{R}^n$

$$\dot{p} = X(p, u), \tag{1}$$

where $X : \mathbb{R}^n \times \mathbb{R}^m \to \mathbb{R}^n$ and $X(p_*, 0) = 0$. Assume we are given a stabilizing feedback law $u = k(p)$ along with a Lyapunov function $V : \mathbb{R}^n \to \mathbb{R}$ that serves as a certificate of the asymptotic stability of the equilibrium $p_* \in \mathbb{R}^n$ under the closed-loop system. Formally,

$$\dot{V} = \langle \nabla V(p), X(p, k(p)) \rangle \leq -F(p), \tag{2}$$

with $F$ a positive definite function relative to $p_*$. For simplicity, we restrict ourselves to the case $F(p) = \alpha V(p)$, with $\alpha \in \mathbb{R}_{>0}$ (in this case, the convergence of $V$ is exponential). The controller $u = k(p)$ cannot be implemented in real time, because it requires both continuous sampling and actuation. The real-time implementation of the closed-loop system can be tackled by considering a sample-and-hold implementation of (1) of the form

$$\dot{p} = X(p, k(\hat{p})), \tag{3}$$

with $p(0) = \hat{p}$, where $\hat{p}$ is a sampled version of the state $p$. The most common approach consists of periodically sampling the state, selecting a stepsize small enough to ensure that the function $V$ remains monotonically decreasing for the resulting system. However, constant stepsizes are generally conservative, as they need to deal with worst-case scenarios. Instead, opportunistic state-triggered control seeks to adjust the stepsize as determined by the current system state. Formally, let $\{t_1, t_2, \ldots\}$ be a sequence of triggering times and denote $p_i = p(t_i)$, for $i \in \mathbb{N}$. Consider

$$\dot{p} = X(p, k(p_i)), \quad \text{for } t \in [t_i, t_{i+1}] \text{ and } i \in \mathbb{N}. \tag{4}$$

The objective is then to identify a criterion to select the sequence of triggering times in a way that ensures that (i) the triggered dynamics (4) retains the guarantees on the evolution of the Lyapunov function and (ii) the inter-sampling times are lower bounded. Condition (ii) ensures feasibility and rules out the possibility of *Zeno* behavior, cf. [9], whereas condition (i) ensures that the triggered dynamics has the same convergence properties as the original dynamics.

Interestingly, both conditions can be met with designs that involve the Lyapunov function $V$ itself. *Event-triggered designs* compute the sequence of triggering times by monitoring the evolution of certain function until a condition is violated. More precisely, assume that we have access to a continuous function $g : \mathbb{R}^n \times \mathbb{R} \to \mathbb{R}$ that satisfies $g(p, 0) < 0$ for all $p \in \mathbb{R}^n \setminus \{p_*\}$ and

$$\dot{V}(p(t)) + \alpha V(p(t)) \leq g(\hat{p}, t),$$

holds along the solutions of (3). Then, for each $i \in \mathbb{N}$, the next triggering time can be determined by

$$t_{i+1} = \min\{t \mid t > t_i \text{ such that } g(p_i, t) = 0\}.$$

Note that, by design, this choice ensures that $\dot{V}(p(t)) \leq -\alpha V(p(t))$ along the dynamics (4). If $g$ is such that $t_{i+1}$, as defined above, can be determined explicitly only with knowledge of $p_i$, one refers to this design as *self-triggered* (because it does not require the continuous monitoring of the evolution of the state under (3) in order to identify it).

## 2.3 Adaptive-Stepsize Forward-Euler Discretization of Continuous-Time Dynamics via Opportunistic State Triggering

The ideas described in Section 2.2 can also be applied in the context of discretization of asymptotically stable continuous-time dynamical systems, as we explain next. Consider a dynamical system on $\mathbb{R}^n$,

$$\dot{p} = Y(p) \tag{5}$$

where $Y : \mathbb{R}^n \to \mathbb{R}^n$. Assume $p_*$ is a globally asymptotically stable equilibrium point under this dynamics, and a certificate, in the form of a Lyapunov function $V : \mathbb{R}^n \to \mathbb{R}$, is available, meaning that $\dot{V} = \langle \nabla V(p), Y(p) \rangle \leq -\alpha V(p)$ for all $p \in \mathbb{R}^n$. Following the state-triggered approach described above, consider the sampled implementation of the dynamics described by

$$\dot{p} = Y(\hat{p}), \tag{6}$$

with $p(0) = \hat{p}$. Note that, the righthand side being constant, this is equivalent to writing

$$p(t) = \hat{p} + tY(\hat{p}), \tag{7}$$

which exactly corresponds to a forward-Euler discretization of stepsize $t$. Therefore, a successful opportunistic state-triggered design would ensure that the monotonic behavior of the Lyapunov function is respected, in turn guaranteeing convergence to the equilibrium at the same rate as the original dynamics (5). Given the connection noted above with the Euler discretization, such state-triggered implementation admits an interesting interpretation from a numerical viewpoint, cf. Figure 1. In fact, the state-triggered implementation exactly corresponds to a variable stepsize Euler discretization where, at each iterate, the trigger criteria helps us determine the stepsize according to the decay criteria specified by the Lyapunov function. Before this decay condition is violated, the state is re-sampled, and the process is repeated. By design, the resulting variable-stepsize Euler discretization retains the convergence rate of the original dynamics.

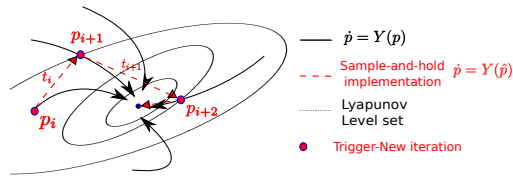

Figure 1: Equivalence between opportunistic state-triggered implementation and variable-stepsize forward-Euler discretization. The black lines correspond to the trajectories of the original dynamics (5). The red lines are trajectories of the family of sampled dynamical systems (6), which are the same as the iterates of forward Euler methods with different stepsizes.

We finish this section by pointing out that continuous models of accelerated optimization algorithms, particularly high-resolution ODEs, fit the profile described above (i.e., they are globally asymptotically stable and their convergence can be characterized via suitable Lyapunov functions). Furthermore, their acceleration properties are explained as a consequence of the decay rate of the Lyapunov function. This matches perfectly with our state-triggered approach, which seeks to conserve the decay rate of the Lyapunov function, consequently ensuring the acceleration properties in the resulting discrete-time algorithm. An interesting challenge arises because of the fact that the Lyapunov function typically relies on knowledge of the optimizer, thereby complicating the evaluation of trigger designs based on them. The rest of this paper shows how we tackle this problem to synthesize trigger designs that do not rely on such knowledge and still guarantee the desired decay rate of the Lyapunov function.

# 3 Triggered Discretization of Heavy Ball and Nesterov's Continuous Models

Here we present discretizations, using the methodology described in Section 2.3, of the high-resolution ordinary differential equations (heavy-ball and Nesterov for strongly convex functions) proposed in [23] for optimization. Due to space constraints, we discuss in detail the heavy-ball case and refer the reader to the supplementary material for an analogous discretization of the Nesterov's accelerated gradient for strongly convex functions.

Let $f$ belong to $\mathcal{S}_{\mu,L}^1(\mathbb{R}^n)$ and let $x_*$ be its unique minimizer. Given $s \in \mathbb{R}_{>0}$, consider the following $s$-dependent family of second-order differential equations,

$$\begin{bmatrix} \dot{x} \\ \dot{v} \end{bmatrix} = \begin{bmatrix} v \\ -2\sqrt{\mu}v - (1 + \sqrt{\mu s})\nabla f(x)) \end{bmatrix},$$ (8)

with initial conditions $x(0) = x_0$, $v(0) = -\frac{2\sqrt{s}\nabla f(x_0)}{1+\sqrt{\mu s}}$. When convenient, we refer to this dynamics as $X_{\mathrm{hb}}$. The following result characterizes the convergence properties of this dynamics.

**Theorem 3.1** ([23]). *Let $V : \mathbb{R}^n \times \mathbb{R}^n \to \mathbb{R}$ be the positive definite function relative to $[x_*, 0]^T$,*

$$V(x,v) = (1 + \sqrt{\mu s})(f(x) - f(x_*)) + \frac{1}{4}\|v\|^2 + \frac{1}{4}\|v + 2\sqrt{\mu}(x - x_*)\|^2.$$

*Then $\dot{V} \leq -\frac{\sqrt{\mu}}{4}V$ along the dynamics (8) and, as a consequence, $[x_*, 0]^T$ is globally asymptotically stable. Moreover, for $s \leq 1/L$, the decay rate of the Lyapunov function $V$ implies*

$$f(x(t)) - f(x_*) \leq \frac{7\|x(0) - x_*\|^2}{2s}e^{-\frac{\sqrt{\mu}}{4}t}.$$ (9)

Given our discussion in Section 2.2, Theorem 3.1 provides all the necessary ingredients to develop a discretization that respects the convergence rate, and hence inherits the guarantee (9). For simplicity, we use the shorthand notation $p = [x, v]^T$. Observe that the Lyapunov function $V$ depends on the minimizer, $x_*$, which is unknown. To circumvent this issue, we resort to tight estimates provided by the convexity properties of the function $f$.

Consider the sampled-and-hold implementation of (8) given by $\dot{p} = X_{\mathrm{hb}}(\hat{p})$, $p(0) = \hat{p}$ or, equivalently, $p(t) = \hat{p} + tX_{\mathrm{hb}}(\hat{p})$. Our goal is to determine a stepsize $t$, as large as possible, that guarantees $\frac{d}{dt}V(p(t)) + \frac{\sqrt{\mu}}{4}V(p(t)) \leq 0$ along the sampled dynamics. The following result provides us with a particularly useful upper bound to ensure this. The proof is provided in the supplementary material.

**Proposition 3.2.** *For the sample-and-hold dynamics $\dot{p} = X_{\mathrm{hb}}(\hat{p})$, $p(0) = \hat{p}$, $0 \leq s$ and $0 \leq \alpha \leq \sqrt{\mu}/4$. Let*

$$\frac{d}{dt}V(p(t)) + \alpha V(p(t)) = \langle \nabla V(\hat{p} + tX_{\mathrm{hb}}(\hat{p})), X_{\mathrm{hb}}(\hat{p})\rangle + \alpha V(\hat{p} + tX_{\mathrm{hb}}(\hat{p}))$$

$$= \underbrace{\langle \nabla V(\hat{p} + tX_{\mathrm{hb}}(\hat{p})) - \nabla V(\hat{p}), X_{\mathrm{hb}}(\hat{p})\rangle}_{I} + \underbrace{\alpha(V(\hat{p} + tX_{\mathrm{hb}}(\hat{p})) - V(\hat{p}))}_{II}$$

$$+ \underbrace{\langle \nabla V(\hat{p}), X_{\mathrm{hb}}(\hat{p})\rangle + \alpha V(\hat{p})}_{III}.$$

*Then, the following bounds hold:*

1. *Term $I \leq A_{\mathrm{ET}}(\hat{p}, t) \leq A_{\mathrm{ST}}(\hat{p})t$;*

2. *Term $II \leq BC_{\mathrm{ET}}(\hat{p}, t) \leq B_{\mathrm{ST}}(\hat{p})t + C_{\mathrm{ST}}(\hat{p})t^2$;*

3. *Term $III \leq D_{\mathrm{ET}}(\hat{p}, t) = D_{\mathrm{ST}}(\hat{p})$,*

*where*

$$A_{\mathrm{ET}}(\hat{p}, t) = (1 + \sqrt{\mu s})\langle \nabla f(\hat{x} + t\hat{v}) - \nabla f(\hat{x}), \hat{v}\rangle + t2\sqrt{\mu}(1 + \sqrt{\mu s})\langle \nabla f(\hat{x}), \hat{v}\rangle + t2\mu \|\hat{v}\|^2$$
$$+ t(1 + \sqrt{\mu s})^2 \|\nabla f(\hat{x})\|^2,$$

$$BC_{\mathrm{ET}}(\hat{p}, t) = \alpha\big((1 + \sqrt{\mu s})(f(\hat{x} + t\hat{v}) - f(\hat{x})) + t^2\frac{1}{4}\|-2\sqrt{\mu}\hat{v} - (1 + \sqrt{\mu s})\nabla f(\hat{x})\|^2$$
$$- t(1 + \sqrt{\mu s})\langle \hat{v}, \nabla f(\hat{x})\rangle - t\sqrt{\mu}\|\hat{v}\|^2 + t^2\frac{1}{4}\|(1 + \sqrt{\mu s})\nabla f(\hat{x})\|^2$$
$$- t\sqrt{\mu}(1 + \sqrt{\mu})\|\nabla f(\hat{x})\|^2/L),$$

$$D_{\mathrm{ET}}(\hat{p}, t) = (\alpha\frac{3}{4} - \sqrt{\mu})\|\hat{v}\|^2 + \big((1 + \sqrt{\mu s})\frac{\alpha - \sqrt{\mu}}{2L} + (\alpha 2\mu - \frac{\sqrt{\mu}(1 + \sqrt{\mu s})\mu)}{2})\frac{1}{L^2}\big)\|\nabla f(\hat{x})\|^2,$$

$$A_{\mathrm{ST}}(\hat{p}) = (1 + \sqrt{\mu s})L\|\hat{v}\|^2 + 2\sqrt{\mu}(1 + \sqrt{\mu s})\langle \nabla f(\hat{x}), \hat{v}\rangle + 2\mu\|\hat{v}\|^2 + (1 + \sqrt{\mu s})^2\|\nabla f(\hat{x})\|^2,$$

$$B_{\mathrm{ST}}(\hat{p}) = \alpha\big(-\sqrt{\mu}\|\hat{v}\|^2 - \sqrt{\mu}(1 + \sqrt{\mu s})\frac{1}{L}\|\nabla f(\hat{x})\|^2\big),$$

$$C_{\mathrm{ST}}(\hat{p}) = \alpha\big((1 + \sqrt{\mu s})\frac{L}{2}\|\hat{v}\|^2 + \frac{1}{4}\|-2\sqrt{\mu}\hat{v} - (1 + \sqrt{\mu s})\nabla f(\hat{x})\|^2 + \frac{1}{4}\|-(1 + \sqrt{\mu s})\nabla f(\hat{x})\|^2\big).$$

We define, with a slight abuse of notation,

$$g_{\mathrm{ET}}(\hat{p}, t) = A_{\mathrm{ET}}(\hat{p}, t) + BC_{\mathrm{ET}}(\hat{p}, t) + D_{\mathrm{ET}}(\hat{p}, t),$$
$$g_{\mathrm{ST}}(\hat{p}, t) = C_{\mathrm{ST}}(\hat{p})t^2 + (A_{\mathrm{ST}}(\hat{p}) + B_{\mathrm{ST}}(\hat{p}))t + D_{\mathrm{ST}}(\hat{p}),$$

(the reason for the subindexes ET, for event-triggered, and ST, for self-triggered, becomes clear below). With these functions in place, it follows from Proposition 3.2 that

$$\frac{d}{dt}V(p(t)) + \alpha V(p(t)) \leq g_{\mathrm{ET}}(\hat{p}, t) \leq g_{\mathrm{ST}}(\hat{p}, t). \tag{10}$$

This is all we need to determine the stepsize starting from $\hat{p}$. Formally, we set

$$\mathrm{step}_{\#}(\hat{p}) = \min_t\{t > 0 \text{ such that } g_{\#}(\hat{p}, t) = 0\}, \tag{11}$$

where $\# \in \{\mathrm{ET}, \mathrm{ST}\}$. Note that, when $\# = \mathrm{ET}$, then $g_{\mathrm{ET}}(\hat{p}, t) = 0$ is an implicit equation on $t$. Instead, when $\# = \mathrm{ST}$, then the solution to the quadratic equation $g_{\mathrm{ST}}(\hat{p}, t) = 0$ can be obtained explicitly (i.e., determined only with the information about the current state $\hat{p}$) since $C_{\mathrm{ST}}(\hat{p}) > 0$ and $D_{\mathrm{ST}}(\hat{p}) < 0$ when $\alpha \leq \sqrt{\mu}/4$. In fact, we have

$$\mathrm{step}_{\mathrm{ST}}(\hat{p}) = \frac{-(A_{\mathrm{ST}}(\hat{p}) + B_{\mathrm{ST}}(\hat{p})) + \sqrt{(A_{\mathrm{ST}}(\hat{p}) + B_{\mathrm{ST}}(\hat{p}))^2 - 4C_{\mathrm{ST}}(\hat{p})D_{\mathrm{ST}}(\hat{p})}}{2C_{\mathrm{ST}}(\hat{p})}.$$

Algorithm 1 describes in pseudocode the resulting variable-stepsize integrator.

---

**Algorithm 1:** Triggered Forward-Euler algorithm

---
**Initialization:** Initial point ($p_0$), convergence rate ($\alpha$), objective function ($f$), tolerance ($\epsilon$);
**Set:** $k = 0$;
**while** $\|\nabla f(x)\| \geq \epsilon$ **do**
    Compute stepsize $t_k$ at current point according to (11);
    Compute next iterate $p_{k+1} = p_k + t_k X_{\mathrm{hb}}(p_k)$;
    Set $k = k + 1$
**end**

---

**Theorem 3.3.** *For $0 < \alpha \leq \sqrt{\mu}/4$ and $\# \in \{\mathrm{ET}, \mathrm{ST}\}$, Algorithm 1 is a variable-stepsize integrator with the following properties*

    *(i) the stepsize is uniformly lower bounded by a positive constant. Namely*

$$-\bar{c}_2 + \sqrt{\bar{c}_2^2 + \bar{c}_1} \leq \mathrm{step}_{ST}(p)$$

*where*

$$\bar{c}_1 = \min\{\frac{2\left(\sqrt{\mu} - \frac{3\alpha}{4}\right)}{\alpha\left(4\mu + L\sqrt{\mu s} + L\right)}, \frac{2\left(-4\alpha\mu + L\left(\sqrt{\mu} - \alpha\right)\left(\sqrt{\mu s} + 1\right) + \mu^{3/2}\left(\sqrt{\mu s} + 1\right)\right)}{3\alpha L^2\left(\sqrt{\mu s} + 1\right)^2}\}$$

$$\bar{c}_2 = \max\{\frac{\left(2\mu + \sqrt{\mu} + L\right)\left(\sqrt{\mu s} + 1\right)}{\alpha\left(4\mu + L\sqrt{\mu s} + L\right)}, \frac{2\left(\sqrt{\mu} + \sqrt{\mu s} + 1\right)}{3\alpha\left(\sqrt{\mu s} + 1\right)}\}.$$

(ii) $\frac{d}{dt}V(p_k + tX_{\mathrm{hb}}(p_k)) \leq -\alpha V(p_k + tX_{\mathrm{hb}}(p_k))$ *for all* $t \in [0, t_k]$ *and all* $k \in \{0\} \cup \mathbb{N}$.

*As a consequence, it follows that* $f(x_{k+1}) - f(x_*) \leq \frac{7\|x(0) - x_*\|^2}{2s}e^{-\alpha\sum_{i=0}^k t_i}$ *for all* $k \in \{0\} \cup \mathbb{N}$.

*Proof.* Since $g_{\mathrm{ET}}(p, t) \leq g_{\mathrm{ST}}(p, t)$ we have $\mathrm{step}_{\mathrm{ST}}(p) \leq \mathrm{step}_{\mathrm{ET}}(p)$ and therefore it is enough to prove the first claim for the ST-case. We rewrite,

$$\mathrm{step}_{\mathrm{ST}}(p) = \frac{-(A_{\mathrm{ST}}(p) + B_{\mathrm{ST}}(p))}{2C_{\mathrm{ST}}(p)} + \sqrt{\left(\frac{A_{\mathrm{ST}}(p) + B_{\mathrm{ST}}(p)}{2C_{\mathrm{ST}}(p)}\right)^2 - \frac{D_{\mathrm{ST}}(p)}{C_{\mathrm{ST}}(p)}}.$$

We bound, using $\|a + b\|^2 \leq 2\|a\|^2 + 2\|b\|^2$,

$$C_{\mathrm{ST}}(p) \leq \alpha\big(((1 + \sqrt{\mu s})\frac{L}{2} + 2\mu)\|v\|^2 + \frac{3}{4}(1 + \sqrt{\mu s})^2\|\nabla f(x)\|^2\big)$$

and therefore

$$\frac{-D_{\mathrm{ST}}(p)}{C_{\mathrm{ST}}(p)} \geq \frac{-(\alpha\frac{3}{4} - \sqrt{\mu})\|v\|^2 - \big((1 + \sqrt{\mu s})\frac{\alpha - \sqrt{\mu}}{2L} + (\alpha 2\mu - \frac{\sqrt{\mu}(1 + \sqrt{\mu s})\mu)}{2})\frac{1}{L^2}\big)\|\nabla f(x)\|^2}{\alpha\big(((1 + \sqrt{\mu s})\frac{L}{2} + 2\mu)\|v\|^2 + \frac{3}{4}(1 + \sqrt{\mu s})^2\|\nabla f(x)\|^2\big)}.$$

We observe that if we rename $\|\nabla f(x)\| = z_1$ and $\|v\| = z_2$ then the last expression has the form

$$\frac{\beta_1 z_1^2 + \beta_2 z_2^2}{\beta_3 z_1^2 + \beta_4 z_2^2}. \tag{12}$$

We show in the supplementary material that such expression is upper and lower bounded by positive constants, i.e., there exist (explicit) $c_1$ and $c_2 \in \mathbb{R}_{>0}$ such that

$$0 < c_1 \leq \frac{\beta_1 z_1^2 + \beta_2 z_2^2}{\beta_3 z_1^2 + \beta_4 z_2^2} \leq c_2 \quad \text{for all } z_1, z_2 \in \mathbb{R}\backslash\{0\}.$$

Using this observation, we have

$$\frac{-(A_{\mathrm{ST}}(p) + B_{\mathrm{ST}}(p))}{2C_{\mathrm{ST}}(p)} + \sqrt{\left(\frac{A_{\mathrm{ST}}(p) + B_{\mathrm{ST}}(p)}{2C_{\mathrm{ST}}(p)}\right)^2 + c_1}$$

$$\leq \frac{-(A_{\mathrm{ST}}(p) + B_{\mathrm{ST}}(p))}{2C_{\mathrm{ST}}(p)} + \sqrt{\left(\frac{A_{\mathrm{ST}}(p) + B_{\mathrm{ST}}(p)}{2C_{\mathrm{ST}}(p)}\right)^2 - \frac{D_{\mathrm{ST}}(p)}{C_{\mathrm{ST}}(p)}}.$$

It is easy to see that the function $f(z) = -z + \sqrt{z^2 + c_1}$ is monotonically decreasing and positive everywhere. So, if $z$ is upper bounded, then $f(z)$ is lower bounded by a positive constant. With this observation, and the form of the last expression, it is clear that if we upper bound $z = \frac{A_{\mathrm{ST}}(p) + B_{\mathrm{ST}}(p)}{2C_{\mathrm{ST}}(p)}$ we are done. To achieve this goal let us use

$$C_{\mathrm{ST}}(p) \geq \alpha\big((1 + \sqrt{\mu s})\frac{L}{2}\|v\|^2 + \frac{1}{4}(1 + \sqrt{\mu s})^2\|\nabla f(x)\|^2\big).$$

and

$$A_{\mathrm{ST}}(p) + B_{\mathrm{ST}}(p) \leq A_{\mathrm{ST}}(p) \leq (1 + \sqrt{\mu s})L\|v\|^2 + \sqrt{\mu}(1 + \sqrt{\mu s})\|\nabla f(x)\|^2$$
$$+ \sqrt{\mu}(1 + \sqrt{\mu s})\|v\|^2 + 2\mu\|v\|^2 + (1 + \sqrt{\mu s})^2\|\nabla f(x)\|^2$$

where we have used Cauchy-Schwartz and Young's inequality in the last estimate. Now, the fraction $\frac{A_{\mathrm{ST}}(p) + B_{\mathrm{ST}}(p)}{2C_{\mathrm{ST}}(p)}$ has the form (12), and so we can conclude the existence of $c_2$ such that $\frac{A_{\mathrm{ST}}(p) + B_{\mathrm{ST}}(p)}{2C_{\mathrm{ST}}(p)} \leq c_2$. To finish now the proof of the first part of Theorem 3.3 it is only necessary to use the explicit expressions of $c_1$ and $c_2$ provided in the supplementary material. The second part follows from Proposition 3.2 and the algorithm design. $\square$

We compare the performance of Algorithm 1 for the event-triggered (ET) and self-triggered (ST) cases with the explicit and symplectic integrators proposed in [24] in a logistic regression example. Figure 2 illustrates the evolution of the stepsize, objective and Lyapunov functions. We set $\alpha = \sqrt{\mu}/4$ and $s = \mu/(36L^2)$ following the values in [24]. The objective function corresponds to the regularized logistic regression cost function, namely $\sum_{i=1}^{10} \log(1 + e^{-y_i \langle v_i, x \rangle}) + 1/2 \|x\|^2$, where $x \in \mathbb{R}^4$ and we have generated the sampled points $(v_i, y_i)$ randomly. This function is 1-strongly convex. The value of $L = 177.49$ can be estimated by straightforward computations. In the plots, we display the optimal stepsize only for comparison purposes, as the minimizer is in practice unknown. Knowledge of the minimizer $x_*$ would enable the explicit computation of the Lyapunov function, cf. Theorem 3.1, which in turn would allow to solve $\dot{V} + \alpha V$ by any standard numerical method at any iteration. This is what we refer to as optimal stepsize.

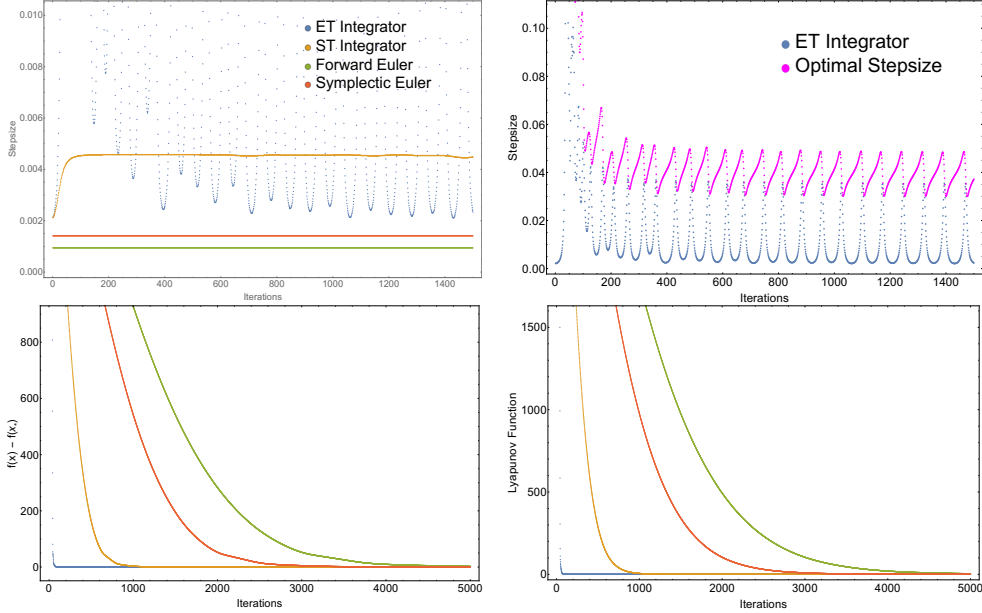

Figure 2: (Top, left) Comparison of stepsizes along the various discrete-time dynamics. The ET and ST integrators keep a larger stepsize for the first 2000 iterations, approaching the optimizer much faster. (Top, right) Comparison of the evolution of the ET stepsize and optimal stepsize along the ET dynamics. We observe how the stepsize computed by the ET integrator chases the optimal stepsize as the state evolves. (Bottom, left) Comparison of the evolution of the objective function along the different dynamics. (Bottom, right) Comparison of the evolution of the Lyapunov function along the different dynamics.

Figure 3 provides another comparison for a quadratic objective function over $\mathbb{R}^{50}$ defined by an (ill-conditioned) positive definite $50 \times 50$ matrix, where $\mu = 3.5$ and $L = 7006.6$. We plot the evolution of the objective and the logarithm of the Lyapunov functions, comparing the proposed algorithms with forward and symplectic Euler. The supplementary material contains additional comparisons for various 2-dimensional quadratic cases.

## 4 Conclusions and Future Work

We have introduced a novel opportunistic state-triggered approach to the discretization of optimization flows. Our approach relies on the key observation that resource-aware control provides a principled way of going from continuous-time control design to real-time implementation with stability and performance guarantees. This is done by opportunistically prescribing when certain action should occur. In this case, the action amounts to a certain decay of the Lyapunov function. The presented framework provides a promising path towards the design of adaptive optimization algorithms. We have provided theoretical guarantees that ensure the implementability of the method and numerical comparisons with recent discretizations of the heavy-ball dynamics. The supplemental material contains analogous results for the Nesterov's accelerated gradient for strongly convex functions.

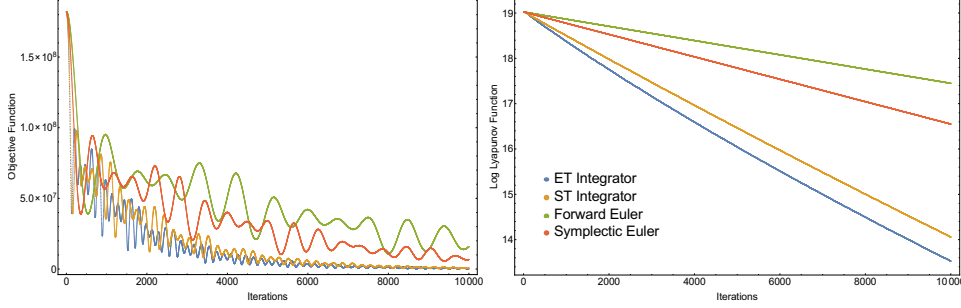

Figure 3: (Left) Comparison of the evolution of the objective function along the different dynamics. (Right) Comparison of the evolution of the logarithm of the Lyapunov function along the different dynamics.

We have employed a derivative-based approach to trigger design combined with the forward Euler method for its simplicity, but believe that other powerful schemes can be synthesized in the future by resorting to the following ideas.

*Use of more complex integrators.* The setting presented here is general enough to incorporate other integrators beyond the forward Euler method that may yield better performance. Additionally, the sampled information employed in our approach is a zero-order-hold, and possibilities exist within the theory of resource-aware control to employ higher-order holds that more accurately approximate the evolution of the continuous-time dynamics. The direct application of the forward Euler method to Nesterov's continuous model gives a dynamics that includes the second-order term $\nabla^2 f(x)v$. This is a drawback, as precisely the success of Nesterov's method is the requirement of only first-order information. Two promising approaches to circumvent this issue are to approximate the term $\sqrt{s}\nabla^2 f(x)v$ by $\nabla f(x_{k+1}) - \nabla f(x_k)$, cf. [19, 24], and to recast the second-order Nesterov's ODE as a first-order one, cf. [2, 3] and develop analogous schemes for the resulting dynamics.

*Convergence rate as a result of Lyapunov decay and uniform lower bound on stepsize.* The result in Theorem 3.3 links the convergence rate of the discrete-time algorithm to the Lyapunov decay and the stepsize of the state-triggered implementation of the continuous-time dynamics. More explicitly, if we bound the stepsize by $\hat{t}$ then $f(x_{k+1}) - f(x_*) \in \mathcal{O}(\exp(-\frac{\sqrt{\mu}}{4}\hat{t})^k)$. Therefore acceleration can be understood as a consequence of the ability of the state-triggered implementation to maintain certain Lyapunov decay for a long enough time (i.e., large stepsize). Although we do not observe acceleration in the numerical studies presented here, i.e., $\exp(-\frac{\sqrt{\mu}}{4}\hat{t}) \geq 1 - \sqrt{\frac{\mu}{L}}$, this is probably due to the simplicity of the used forward-Euler integrator employed. Nonetheless, the introduced variable stepsize integrators clearly outperform their equivalent fixed-stepsize counterparts, reinforcing the importance of extending our design to more complex integrators with which to achieve the desired convergence rates.

*Use of other triggering conditions.* Other approaches to trigger design, beyond derivative-based ones, are promising. For instance, [26] introduces a Lyapunov sampling event-triggered approach whose main idea is to continuously sample the Lyapunov function until certain decay has been reached. The trigger takes the form $t_{k+1} = \min\{t > t_i \text{ such that } V(x(t)) - \eta V(x_i) = 0\}$, where $\eta \in [0, 1]$ is a design parameter. The expression is similar to the difference $V(x_{i+1}) - V(x_i)$, which is upper bounded in [24] for the iterations of explicit and implicit symplectic integrators, and plays a key role in the convergence analysis. This suggests the use of similar bounds to develop variable-stepsize integrators. Along the same lines, the use of dynamic triggers [6] to keep track of how much the Lyapunov function decreases along the evolution is also appealing.

*Extensions to convex functions.* The work [23] presents another high-resolution ODE for the case of Nesterov's method applied to convex functions. The sharp bounds on the evolution of the Lyapunov function provided by strong convexity that we employ in the trigger design do not hold anymore. It is therefore challenging and extremely interesting to develop new ideas to tackle this problem.

## Acknowledgments

This work was supported by NSF Award CNS-1446891 and AFOSR Award FA9550-15-1-0108.

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
