[Supplementary Material]

# Supplementary Material: Convergence-Rate-Matching Discretization of Accelerated Optimization Flows Through Opportunistic State-Triggered Control

**Miguel Vaquero**
Mechanical and Aerospace Engineering
UC San Diego
San Diego, CA 9500
`mivaquerovallinas@ucsd.edu`

**Jorge Cortés**
Mechanical and Aerospace Engineering
UC San Diego
San Diego, CA 9500
`cortes@ucsd.edu`

## Abstract

This is the supplementary material corresponding to the paper entitled *"Convergence-Rate-Matching Discretization of Accelerated Optimization Flows Through Opportunistic State-Triggered Control"*. In the first part of this supplement, we provide proofs of some auxiliary results for the heavy-ball case. In the second part, we introduce the continuous model of Nesterov's accelerated gradient method for strongly convex functions presented in [1] and its discretization. Finally, we provide numerical illustrations of the performance of the proposed integratros on several quadratic objective functions.

## 1 Heavy-Ball Case

**Proof of Proposition 3.2**

Here we upper bound the terms I, II and III described in Proposition 3.2 in order to obtain the definitions of $A_{ET}$, $BC_{ET}$, $D_{ET}$, $A_{ST}$, $B_{ST}$, $C_{ST}$ and $D_{ST}$. We will use the notation $p = [x, v]^T$ along this section.

• **Term** $I$. By direct computation we have

$$\nabla V(p) = \begin{bmatrix} (1 + \sqrt{\mu s})\nabla f(x) + \sqrt{\mu}v + 2\mu(x - x_*) \\ v + \sqrt{\mu}(x - x_*) \end{bmatrix},$$

and

$$\nabla V(p + tX_{\mathrm{hb}}(p)) = \begin{bmatrix} (1 + \sqrt{\mu s})\nabla f(x + tv) + \sqrt{\mu}v - t\sqrt{\mu}(1 + \sqrt{\mu s})\nabla f(x) + 2\mu(x - x_*) \\ v - t\sqrt{\mu}v - t(1 + \sqrt{\mu s})\nabla f(x) + \sqrt{\mu}(x - x_*) \end{bmatrix}.$$

Therefore,

$$\begin{aligned} \langle \nabla V(p + tX_{\mathrm{hb}}(p)) - \nabla V(p), X_{\mathrm{hb}}(p) \rangle &= (1 + \sqrt{\mu s})\langle \nabla f(x + tv) - \nabla f(x), v \rangle \\ &\quad + t2\sqrt{\mu}(1 + \sqrt{\mu s})\langle \nabla f(x), v \rangle + t2\mu \|v\|^2 \\ &\quad + t(1 + \sqrt{\mu s})^2 \|\nabla f(x)\|^2. \end{aligned}$$

The RHS of the last expression is precisely $A_{ET}(p, t)$. We can use the Lipschitz continuity of the gradient to get

$$A_{ST}(p) = (1 + \sqrt{\mu s})L \|v\|^2 + 2\sqrt{\mu}(1 + \sqrt{\mu s})\langle \nabla f(x), v \rangle + 2\mu \|v\|^2 + (1 + \sqrt{\mu s})^2 \|\nabla f(x)\|^2.$$

• **Term** $II$**.** We have

$$V(p + tX_{\text{hb}}(p)) - V(p) = (1 + \sqrt{\mu s})(f(x + tv) - f(x_*)) + \frac{1}{4}\left\| v - t2\sqrt{\mu}v - t(1 + \sqrt{\mu s})\nabla f(x)\right\|^2$$

$$+ \frac{1}{4}\left\| v - t2\sqrt{\mu}v - t(1 + \sqrt{\mu s})\nabla f(x) + 2\sqrt{\mu}(x + tv - x_*)\right\|^2$$

$$- (1 + \sqrt{\mu s})(f(x) - f(x_*)) - \frac{1}{4}\left\| v\right\|^2 - \frac{1}{4}\left\| v + 2\sqrt{\mu}(x - x_*)\right\|^2.$$

Using $\|a + b\|^2 = \|a\|^2 + 2\langle a, b\rangle + \|b\|^2$ on the second and third addends, the last expression becomes

$$(1 + \sqrt{\mu s})(f(x + tv) - f(x)) + \underbrace{\frac{1}{4}\left\| v\right\|^2}_{1} + \frac{1}{4}\left\| -t2\sqrt{\mu}v - t(1 + \sqrt{\mu s})\nabla f(x)\right\|^2$$

$$+ \frac{2}{4}\langle v, -t2\sqrt{\mu}v - t(1 + \sqrt{\mu s})\nabla f(x)\rangle + \underbrace{\frac{1}{4}\left\| v + 2\sqrt{\mu}(x - x_*)\right\|^2}_{2} + \frac{1}{4}\left\| -t(1 + \sqrt{\mu s})\nabla f(x)\right\|^2$$

$$+ \frac{2}{4}\langle v + 2\sqrt{\mu}(x - x_*), -t(1 + \sqrt{\mu s})\nabla f(x)\rangle \underbrace{- \frac{1}{4}\left\| v\right\|^2}_{1} \underbrace{- \frac{1}{4}\left\| v + 2\sqrt{\mu}(x - x_*)\right\|^2}_{2}.$$

Canceling out the corresponding terms (1 and 2) we get

$$(1 + \sqrt{\mu s})(f(x + tv) - f(x)) + \frac{1}{4}\left\| -t2\sqrt{\mu}v - t(1 + \sqrt{\mu s})\nabla f(x)\right\|^2$$

$$+ \frac{2}{4}\langle v, -t2\sqrt{\mu}v - t(1 + \sqrt{\mu s})\nabla f(x)\rangle + \frac{1}{4}\left\| -t(1 + \sqrt{\mu s})\nabla f(x)\right\|^2$$

$$+ \frac{2}{4}\langle v + 2\sqrt{\mu}(x - x_*), -t(1 + \sqrt{\mu s})\nabla f(x)\rangle.$$

Finally, using

$$f(x + tv) - f(x) \le \langle \nabla f(x), tv\rangle + \frac{L}{2}\left\| tv\right\|^2,$$

$$\langle x_* - x, \nabla f(x)\rangle \le -\frac{\|\nabla f(x)\|^2}{L},$$

we can uncover the expression of $BC_{ET}$. We continue to obtain the definitions of $B_{ST}$ and $C_{ST}$

$$(1 + \sqrt{\mu s})(\underbrace{t\langle\nabla f(x), v\rangle}_{1} + t^2\frac{L}{2}\left\| v\right\|^2) + t^2\frac{1}{4}\left\| -2\sqrt{\mu}v - (1 + \sqrt{\mu s})\nabla f(x)\right\|^2$$

$$+ t\frac{1}{2}\langle v, -2\sqrt{\mu}v \underbrace{-(1 + \sqrt{\mu s})\nabla f(x)}_{1}\rangle + t^2\frac{1}{4}\left\| -(1 + \sqrt{\mu s})\nabla f(x)\right\|^2$$

$$\underbrace{- t\frac{1}{2}(1 + \sqrt{\mu s})\langle v, \nabla f(x)\rangle}_{1} - t\sqrt{\mu}(1 + \sqrt{\mu s})\frac{1}{L}\left\| \nabla f(x)\right\|^2.$$

After canceling out the terms with 1 we get

$$t^2(1 + \sqrt{\mu s})\frac{L}{2}\left\| v\right\|^2 + t^2\frac{1}{4}\left\| -2\sqrt{\mu}v - (1 + \sqrt{\mu s})\nabla f(x)\right\|^2 - t\sqrt{\mu}\left\| v\right\|^2$$

$$+ t^2\frac{1}{4}\left\| -(1 + \sqrt{\mu s})\nabla f(x)\right\|^2 - t\sqrt{\mu}(1 + \sqrt{\mu s})\frac{1}{L}\left\| \nabla f(x)\right\|^2,$$

and so, including the $\alpha$ factor

$$B_{ST}(p) = \alpha\left(-\sqrt{\mu}\left\| v\right\|^2 - \sqrt{\mu}(1 + \sqrt{\mu s})\frac{1}{L}\left\| \nabla f(x)\right\|^2\right)),$$

$$C_{ST}(p) = \alpha\left((1 + \sqrt{\mu s})\frac{L}{2}\left\| v\right\|^2 + \frac{1}{4}\left\| -2\sqrt{\mu}v - (1 + \sqrt{\mu s})\nabla f(x)\right\|^2\right.$$

$$\left. + \frac{1}{4}\left\| -(1 + \sqrt{\mu s})\nabla f(x)\right\|^2\right).$$

• **Term** $III$**.** From the computations in Appendix B in [1] (p. 50) we have

$$\langle \nabla V(p), X_{\mathrm{hb}}(p) \rangle \leq -\sqrt{\mu}(\|v\|^2 + (1 + \sqrt{\mu s})\langle \nabla f(x), x - x_* \rangle).$$

On the other hand

$$
\begin{aligned}
\alpha V(p) \;&=\; \alpha\big((1 + \sqrt{\mu s})(f(x) - f(x_*)) + \tfrac{1}{4}\|v\|^2 + \tfrac{1}{4}\left\|v + 2\sqrt{\mu}(x - x_*)\right\|^2\big) \\
&\leq \alpha\big((1 + \sqrt{\mu s})(f(x) - f(x_*)) + \tfrac{1}{4}\|v\|^2 + \tfrac{2}{4}\|v\|^2 + \tfrac{2}{4}\|2\sqrt{\mu}(x - x_*)\|^2\big),
\end{aligned}
$$

where we used $\|a + b\|^2 \leq 2\|a\|^2 + 2\|b\|^2$ on the last addend. Finally, using

$$\langle \nabla f(x), x_* - x \rangle \leq f(x_*) - f(x) - \tfrac{\mu}{2}\|x - x_*\|^2,$$

$$f(x_*) - f(x) \leq -\frac{\|\nabla f(x)\|^2}{2L},$$

$$-\|x - x_*\|^2 \leq \frac{-\|\nabla f(x)\|^2}{L^2},$$

if $0 \leq \alpha \leq \frac{\sqrt{\mu}}{4}$, it is easy to see that

$$
\begin{aligned}
\langle \nabla V(p), X_{\mathrm{hb}}(p) \rangle + \alpha V(p) \;&\leq\; (\alpha\tfrac{3}{4} - \sqrt{\mu})\|v\|^2 \\
&\quad + \big((1 + \sqrt{\mu s})\tfrac{\alpha - \sqrt{\mu}}{2L} + (\alpha 2\mu - \tfrac{\sqrt{\mu}(1 + \sqrt{\mu s})\mu)}{2})\tfrac{1}{L^2}\big)\|\nabla f(x)\|^2.
\end{aligned}
$$

And so,

$$D_{ET}(p) = D_{ST}(p) = (\alpha\frac{3}{4} - \sqrt{\mu})\|v\|^2 + \big((1 + \sqrt{\mu s})\frac{\alpha - \sqrt{\mu}}{2L} + (\alpha 2\mu - \frac{\sqrt{\mu}(1 + \sqrt{\mu s})\mu)}{2})\frac{1}{L^2}\big)\|\nabla f(x)\|^2.$$

Which finishes the proof of Proposition 3.2. $\qquad\square$

## Explicit Computation of Uniform Lower Bound of Stepsize

In this section we provide a lower bound of the stepsize computed in Theorem 3.3. Before that, we need to introduce the following lemma

**Lemma 1.** *Given the expression below, where $\beta_i$ are positive real numbers,*

$$\frac{\beta_1 z_1^2 + \beta_2 z_2^2}{\beta_3 z_1^2 + \beta_4 z_2^2},$$

*there exist positive constants given by*

$$c_1 = \min\{\frac{\beta_2}{\beta_4}, \frac{\beta_1}{\beta_3}\}, \quad c_2 = \max\{\frac{\beta_2}{\beta_4}, \frac{\beta_1}{\beta_3}\}. \tag{1}$$

*such that*

$$c_1 \leq \frac{\beta_1 z_1^2 + \beta_2 z_2^2}{\beta_3 z_1^2 + \beta_4 z_2^2} \leq c_2 \quad \text{for all } z_1, \; z_2 \in \mathbb{R}\backslash\{0\}.$$

*Proof.* Consider

$$h(z_1, z_2) = \frac{\beta_1 z_1^2 + \beta_2 z_2^2}{\beta_3 z_1^2 + \beta_4 z_2^2}$$

when $z_2 = 0$ we have $h(z_1, 0) = \frac{\beta_1}{\beta_3}$. When $z_2 \neq 0$ we can divide numerator and denominator by $z_2^2$ and study the following function of $z = \frac{z_1}{z_2}$

$$g(z) = \frac{\beta_1 z^2 + \beta_2}{\beta_3 z^2 + \beta_4}.$$

If we compute the derivative, to compute the minimum and maximum, we have

$$g'(z) = \frac{2z\beta_1(\beta_3 z^2 + \beta_4) - 2\beta_3 z(\beta_1 z^2 + \beta_2)}{(\beta_3 z^2 + \beta_4)^2}$$

and $g'(z) = 0 \Leftrightarrow 2z(\beta_1\beta_4 - \beta_3\beta_2) = 0$. Since the function is symmetric with respect to vertical-axis we only need to study its maximum and minimum between $[0, \infty)$. In this interval $g$ is increasing if $(\beta_1\beta_4 - \beta_3\beta_2) > 0$ and decreasing otherwise. If $(\beta_1\beta_4 - \beta_3\beta_2) = 0$ the function is constant and the maximum and minimum coincide with $\frac{\beta_2}{\beta_4}$. Let us assume that $g$ is increasing, the other case is analogous. Then the maximum is reached asymptotically $\lim_{z \to \infty} g(z) = \frac{\beta_1}{\beta_3}$. The minimum is achieved at $z = 0$, which is $\frac{\beta_2}{\beta_4}$. So $c_1 = \frac{\beta_2}{\beta_4}$ and $c_2 = \frac{\beta_1}{\beta_3}$. In the general case we can easily conclude the desired result.

$\square$

*Computation of a lower bound of the stepsize.* Using Lemma 1 we can compute $c_1$ such that

$$0 < c_1 \leq \frac{-D_{ST}(p)}{C_{ST}(p)}.$$

Taking into account that

$$\frac{-D_{ST}(p)}{C_{ST}(p)} \geq \frac{-(\alpha\frac{3}{4} - \sqrt{\mu})\|v\|^2 - ((1 + \sqrt{\mu s})\frac{\alpha - \sqrt{\mu}}{2L} + (\alpha 2\mu - \frac{\sqrt{\mu}(1+\sqrt{\mu s})\mu)}{2})\frac{1}{L^2})\|\nabla f(x)\|^2}{\alpha((1 + \sqrt{\mu s})\frac{L}{2} + 2\mu)\|v\|^2 + \alpha\frac{3}{4}(1 + \sqrt{\mu s})^2\|\nabla f(x)\|^2},$$

we obtain

$$c_1 = \min\{\frac{2\left(\sqrt{\mu} - \frac{3\alpha}{4}\right)}{\alpha\left(4\mu + L\sqrt{\mu s} + L\right)}, \frac{2\left(-4\alpha\mu + L\left(\sqrt{\mu} - \alpha\right)\left(\sqrt{\mu s} + 1\right) + \mu^{3/2}\left(\sqrt{\mu s} + 1\right)\right)}{3\alpha L^2\left(\sqrt{\mu s} + 1\right)^2}\}.$$

Analogously, let us compute $c_2$ such that

$$\frac{A_{ST}(p) + B_{ST}(p)}{2C_{ST}(p)} \leq c_2,$$

using

$$\frac{A_{ST}(p) + B_{ST}(p)}{2C_{ST}(p)} \leq \frac{A_{ST}(p)}{2C_{ST}(p)} \leq \frac{(1 + \sqrt{\mu s})(L + \sqrt{\mu} + 2\mu)\|v\|^2 + (\sqrt{\mu} + (1 + \sqrt{\mu s}))(1 + \sqrt{\mu s})\|\nabla f(x)\|^2}{2\alpha((1 + \sqrt{\mu s})\frac{L}{2} + 2\mu)\|v\|^2 + 2\alpha\frac{3}{4}(1 + \sqrt{\mu s})^2\|\nabla f(x)\|^2}.$$

Following the same strategy as before,

$$c_2 = \max\{\frac{\left(2\mu + \sqrt{\mu} + L\right)\left(\sqrt{\mu s} + 1\right)}{\alpha\left(4\mu + L\sqrt{\mu s} + L\right)}, \frac{2\left(\sqrt{\mu} + \sqrt{\mu s} + 1\right)}{3\alpha\left(\sqrt{\mu s} + 1\right)}\}.$$

By the arguments introduced in the paper, we can conclude that if $f(z) = -z + \sqrt{z^2 + c_1}$ then

$$f(c_2) \leq \text{step}_{ST}(p),$$

for all $p$. Finally,

$$f(c_2) = -c_2 + \sqrt{c_2^2 + c_1} \leq \text{step}_{ST}(p).$$

## 2 Nesterov's Accelerated Gradient for Strongly Convex Functions

**Continuous Model**

This section is devoted to the introduction and discretization of the continuous model of Nesterov's accelerated gradient for strongly convex functions presented in [1]. Due to the similarities with the heavy-ball case we will only sketch the proofs. Let $f$ be an objective function, $f \in \mathcal{S}_{\mu,L}^2(\mathbb{R}^n)$, and $x_*$ its unique minimizer. The Hessian of $f$ is assumed to be $H$-Lipschitz continuous, so $\|\nabla^2 f(x) - \nabla^2 f(y)\| \leq H\|x - y\|$ for all $x, y \in \mathbb{R}^n$. We also assume knowledge of a bound of the type $\|\nabla^2 f(x)\| \leq M$ for all $x \in \mathbb{R}^n$. Given $s \in \mathbb{R}_{>0}$, consider the following $s$-dependent family of second-order differential equations,

$$\begin{bmatrix} \dot{x} \\ \dot{v} \end{bmatrix} = \begin{bmatrix} v \\ -2\sqrt{\mu}v - \sqrt{s}\nabla^2 f(x)v - (1 + \sqrt{\mu s})\nabla f(x) \end{bmatrix}, \tag{2}$$

with initial conditions $x(0) = x_0$, $v(0) = -\frac{2\sqrt{s}\nabla f(x_0)}{1 + \sqrt{\mu s}}$. When convenient, we refer to this dynamics as $X_{NA}$. The convergence properties of this dynamics are characterized in the next result.

**Theorem 2** ([1]). *Let $V : \mathbb{R}^n \times \mathbb{R}^n \to \mathbb{R}$ be the positive definite function relative to $[x_*, 0]^T$,*

$$V = (1 + \sqrt{\mu s})(f(x) - f(x_*)) + \frac{1}{4}\|v\|^2 + \frac{1}{4}\|v + 2\sqrt{\mu}(x - x_*) + \sqrt{s}\nabla f(x)\|^2. \qquad (3)$$

*Then*

$$\dot{V} \le -\frac{\sqrt{\mu}}{4}V - \frac{\sqrt{s}}{2}(\|\nabla f(x)\|^2 + v^T \nabla^2 f(x)v),$$

*along the dynamics* (2) *and, as a consequence, $[x_*, 0]^T$ is globally asymptotically stable. Moreover, for $s \le 1/L$, the decay rate of the Lyapunov function $V$ implies*

$$f(x(t)) - f(x_*) \le \frac{2\|x(0) - x_*\|^2}{s}e^{-\frac{\sqrt{\mu}}{4}t}, \qquad (4)$$

Analogously to the heavy-ball case, we evaluate $\dot{V} + \alpha V + F$ along the proposed dynamics in order to compute a large stepsize satisfying the decay condition. Let $p(t) = \hat{p} + tX_{\mathrm{NA}}(\hat{p})$, then

$$
\begin{aligned}
\frac{d}{dt}V(p(t)) + \alpha V(p(t)) + F(p(t)) &= \langle \nabla V(\hat{p} + tX_{\mathrm{NA}}(\hat{p})), X_{\mathrm{NA}}(\hat{p})\rangle + \alpha V(\hat{p} + tX_{\mathrm{NA}}(\hat{p})) + F(\hat{p} + tX_{\mathrm{NA}}(\hat{p})) \\
&= \underbrace{\langle \nabla V(\hat{p} + tX_{\mathrm{NA}}(\hat{p})) - \nabla V(\hat{p}), X_{\mathrm{NA}}(\hat{p})\rangle}_{I} + \underbrace{\alpha(V(\hat{p} + tX_{\mathrm{NA}}(\hat{p})) - V(\hat{p}))}_{II} \\
&\quad + \underbrace{F(\hat{p} + tX_{\mathrm{NA}}(\hat{p})) - F(\hat{p})}_{III} + \underbrace{\langle \nabla V(\hat{p}), X_{\mathrm{NA}}(\hat{p})\rangle + \alpha V(\hat{p}) + F(\hat{p})}_{IV}.
\end{aligned}
$$
$$(5)$$

We bound these terms separately. Since our aim here is just showing that analogous results to the heavy-ball case can be obtained, we sketch how to compute the necessary estimates for the self-triggered implementation (*ST*). Event-triggered implementations (*ET*) can be inferred from the computations presented here.

## Upper Bounds of the Terms I, II, III and IV

During the computations of the bounds, we assume $p = [x, v]^T$ is an arbitrary point in $\mathbb{R}^{2n}$.

- **Term $I$.** We have,

$$
\begin{aligned}
\nabla_x V(p) &= (1 + \sqrt{\mu s})\nabla f(x) + \sqrt{\mu}v + 2\mu(x - x_*) + \sqrt{\mu s}\nabla f(x) + \tfrac{\sqrt{s}}{2}\nabla^2 f(x)v \\
&\quad + \sqrt{\mu s}\nabla^2 f(x)(x - x_*) + \tfrac{s}{2}\nabla^2 f(x)\nabla f(x), \\
\nabla_v V(p) &= v + \sqrt{\mu}(x - x_*) + \tfrac{\sqrt{s}}{2}\nabla f(x),
\end{aligned}
$$

and

$$
\begin{aligned}
\nabla_x V(p + tX_{\mathrm{NA}}(p)) &= (1 + \sqrt{\mu s})\nabla f(x + tv) + \sqrt{\mu}(v - t2\sqrt{\mu}v - t\sqrt{s}\nabla^2 f(x)v - t(1 + \sqrt{\mu s})\nabla f(x)) \\
&\quad + 2\mu(x + tv - x_*) + \sqrt{\mu s}\nabla f(x + tv) + \tfrac{\sqrt{s}}{2}\nabla^2 f(x + tv)(v - t2\sqrt{\mu}v - t\sqrt{s}\nabla^2 f(x)v \\
&\quad - t(1 + \sqrt{\mu s})\nabla f(x)) + \sqrt{\mu s}\nabla^2 f(x + tv)(x + tv - x_*) + \tfrac{s}{2}\nabla^2 f(x + tv)\nabla f(x + tv), \\
\nabla_v V(p + tX_{\mathrm{NA}}(p)) &= v - t\sqrt{\mu}v - t\sqrt{s}\nabla^2 f(x)v - t(1 + \sqrt{\mu s})\nabla f(x) + \sqrt{\mu}(x - x_*) + \tfrac{\sqrt{s}}{2}\nabla f(x + tv).
\end{aligned}
$$

After arranging the terms we have

$$\langle \nabla V(p + tX_{\mathrm{NA}}(p)) - \nabla V(p), X_{\mathrm{NA}}(p)\rangle = (1 + \sqrt{\mu s})\langle \nabla f(x + tv) - \nabla f(x), v\rangle$$

$$+ \sqrt{\mu}\langle -t2\sqrt{\mu}v - t\sqrt{s}\nabla^2 f(x)v - t(1 + \sqrt{\mu s})\nabla f(x), v\rangle + 2\mu t\|v\|^2 + \sqrt{\mu s}\langle \nabla f(x + tv) - \nabla f(x), v\rangle$$

$$+ \tfrac{\sqrt{s}}{2}v^T(\nabla^2 f(x + tv) - \nabla^2 f(x))v + \tfrac{\sqrt{s}}{2}v^T \nabla^2 f(x + tv)(-t2\sqrt{\mu}v - t\sqrt{s}\nabla^2 f(x)v - t(1 + \sqrt{\mu s})\nabla f(x))$$

$$+ \sqrt{\mu s}v^T(\nabla^2 f(x + tv) - \nabla^2 f(x))(x - x_*) + \sqrt{\mu s}v^T \nabla^2 f(x + tv)tv + \tfrac{s}{2}v^T \nabla^2 f(x + tv)\nabla f(x + tv)$$

$$- \tfrac{s}{2}v^T \nabla^2 f(x)\nabla f(x) + 2t\mu\|v\|^2 + t\sqrt{\mu s}v^T \nabla^2 f(x)v + t(1 + \sqrt{\mu s})\sqrt{\mu}\langle v, \nabla f(x)\rangle$$

$$+ 2t\sqrt{\mu s}v^T \nabla^2 f(x)v + ts\|\nabla^2 f(x)v\|^2 + t(1 + \sqrt{\mu s})\sqrt{s}\nabla f(x)^T \nabla^2 f(x)v + 2t\sqrt{\mu}(1 + \sqrt{\mu s})\langle \nabla f(x), v\rangle$$

$$+ t(1 + \sqrt{\mu s})\sqrt{s}\nabla f(x)^T \nabla^2 f(x)v + t(1 + \sqrt{\mu s})^2\|\nabla f(x)\|^2 - \sqrt{\mu s}\langle \nabla f(x + tv) - \nabla f(x), v\rangle$$

$$- \tfrac{s}{2}\langle \nabla f(x + tv) - \nabla f(x), \nabla^2 f(x)v\rangle - \tfrac{\sqrt{s}}{2}(1 + \sqrt{\mu s})\langle \nabla f(x + tv) - \nabla f(x), \nabla f(x)\rangle.$$

We cancel out the terms 1, 2, 3, 4, 5 and group the terms with 6

$$(1+\sqrt{\mu s})\langle\nabla f(x+tv)-\nabla f(x),v\rangle + \sqrt{\mu}\langle\underbrace{-t2\sqrt{\mu}v}_{1}\underbrace{-t\sqrt{s}\nabla^2 f(x)v}_{2}\underbrace{-t(1+\sqrt{\mu s})\nabla f(x)}_{3},v\rangle$$

$$+\underbrace{2\mu t\left\|v\right\|^2}_{1}+\underbrace{\sqrt{\mu s}\langle\nabla f(x+tv)-\nabla f(x),v\rangle}_{4}+\tfrac{\sqrt{s}}{2}v^T(\nabla^2 f(x+tv)-\nabla^2 f(x))v+\tfrac{\sqrt{s}}{2}v^T\nabla^2 f(x+tv)(\underbrace{-t2\sqrt{\mu}v}_{5}$$

$$-t\sqrt{s}\nabla^2 f(x)v-t(1+\sqrt{\mu s})\nabla f(x))+\sqrt{\mu s}v^T(\nabla^2 f(x+tv)-\nabla f(x))(x-x_*)+\underbrace{\sqrt{\mu s}v^T\nabla^2 f(x+tv)tv}_{5}$$

$$+\tfrac{s}{2}v^T\nabla^2 f(x+tv)\nabla f(x+tv)-\tfrac{s}{2}v^T\nabla^2 f(x)\nabla f(x)+2t\mu\left\|v\right\|^2+\underbrace{t\sqrt{\mu s}v^T\nabla^2 f(x)v}_{2}$$

$$+\underbrace{t(1+\sqrt{\mu s})\sqrt{\mu}\langle v,\nabla f(x)\rangle}_{3}+2t\sqrt{\mu s}v^T\nabla^2 f(x)v+ts\left\|\nabla^2 f(x)v\right\|^2+\underbrace{t(1+\sqrt{\mu s})\sqrt{s}\nabla f(x)^T\nabla^2 f(x)v}_{6}$$

$$+2t\sqrt{\mu}(1+\sqrt{\mu s})\langle\nabla f(x),v\rangle+\underbrace{t(1+\sqrt{\mu s})\sqrt{s}\nabla f(x)^T\nabla^2 f(x)v}_{6}+t(1+\sqrt{\mu s})^2\left\|\nabla f(x)\right\|^2$$

$$-\underbrace{\sqrt{\mu s}\langle\nabla f(x+tv)-\nabla f(x),v\rangle}_{4}-\tfrac{s}{2}\langle\nabla f(x+tv)-\nabla f(x),\nabla^2 f(x)v\rangle$$

$$-\tfrac{\sqrt{s}}{2}(1+\sqrt{\mu s})\langle\nabla f(x+tv)-\nabla f(x),\nabla f(x)\rangle.$$

Next, we bound any addend separately using the Lipschitz continuity of the gradient and Hessian, and the bound $\|x-x_*\|\leq\frac{\|\nabla f(x)\|}{\mu}$, in combination with the Cauchy-Schwartz inequality. The bound of the terms $\frac{s}{2}v^T\nabla^2 f(x+tv)\nabla f(x+tv)-\frac{s}{2}v^T\nabla^2 f(x)\nabla f(x)$ is computed adding $v^T\nabla^2 f(x+tv)\nabla f(x)-v^T\nabla^2 f(x+tv)\nabla f(x)$ and grouping terms. Finally, we achieve the expression

$$tL(1+\sqrt{\mu s})\left\|v\right\|^2+\tfrac{\sqrt{s}}{2}tH\left\|v\right\|^3+\tfrac{s}{2}t\left\|v\right\|^2 M^2+t\tfrac{\sqrt{s}}{2}(1+\sqrt{\mu s})M\left\|v\right\|\left\|\nabla f(x)\right\|$$

$$+t\tfrac{\sqrt{\mu s}}{\mu}\left\|v\right\|^2 H\left\|\nabla f(x)\right\|+\tfrac{s}{2}tLM\left\|v\right\|^2+\tfrac{s}{2}tH\left\|v\right\|^2\left\|\nabla f(x)\right\|+2t\mu\left\|v\right\|^2$$

$$+2t\sqrt{\mu s}M\left\|v\right\|^2+tsM^2\left\|v\right\|^2+2t(1+\sqrt{\mu s})M\sqrt{s}\left\|\nabla f(x)\right\|\left\|v\right\|+2t\sqrt{\mu}(1+\sqrt{\mu s})\left\|\nabla f(x)\right\|\left\|v\right\|$$

$$+t(1+\sqrt{\mu s})\left\|\nabla f(x)\right\|^2+\tfrac{s}{2}LM\left\|v\right\|^2 t+\tfrac{\sqrt{s}}{2}(1+\sqrt{\mu s})tL\left\|v\right\|\left\|\nabla f(x)\right\|.$$

Notice that all these terms are linear in $t$ and therefore we can define a function $A_{ST}(p)$ analogous to the heavy-ball case. We observe that all addends in the last expression can be upper bounded by terms of the form $(\gamma_1\left\|v\right\|^2+\gamma_2\left\|\nabla f\right\|^2)t$ (where $\gamma_i\in\mathbb{R}$) using Young's inequality, with the exception of the terms $\frac{\sqrt{s}}{2}tH\left\|v\right\|^3$, $t\frac{\sqrt{\mu s}}{\mu}\left\|v\right\|^2 H\left\|\nabla f(x)\right\|$ and $\frac{s}{2}tH\left\|v\right\|^2\left\|\nabla f(x)\right\|$.

• **Term** $II$. Taking into account

$$V(p)=(1+\sqrt{\mu s})(f(x)-f(x_*))+\frac{1}{4}\left\|v\right\|^2+\frac{1}{4}\left\|v+2\sqrt{\mu}(x-x_*)+\sqrt{s}\nabla f(x)\right\|^2$$

and

$$V(p+tX_{\mathrm{NA}}(p))\quad=(1+\sqrt{\mu s})(f(x+tv)-f(x_*))+\tfrac{1}{4}\left\|v-t2\sqrt{\mu}v-t\sqrt{s}\nabla^2 f(x)v-t(1+\sqrt{\mu s})\nabla f(x)\right\|^2$$

$$+\tfrac{1}{4}\left\|v-t2\sqrt{\mu}v-t\sqrt{s}\nabla^2 f(x)v-t(1+\sqrt{\mu s})\nabla f(x)+2\sqrt{\mu}(x+tv-x_*)+\sqrt{s}\nabla f(x+tv)\right\|^2.$$

Using $\|a + b\|^2 = \|a\|^2 + 2\langle a, b\rangle + \|b\|^2$ we have

$$
\begin{aligned}
V(p + tX_{\text{NA}}(p)) - V(p) \quad &\leq (1 + \sqrt{\mu s})(f(x + tv) - f(x)) + \tfrac{1}{4}\|v\|^2 \\
&+ \tfrac{1}{4}\left\|-t2\sqrt{\mu}v - t\sqrt{s}\nabla^2 f(x)v - t(1 + \sqrt{\mu s})\nabla f(x)\right\|^2 \\
&+ \tfrac{2}{4}\langle v, -t2\sqrt{\mu}v - t\sqrt{s}\nabla^2 f(x)v - t(1 + \sqrt{\mu s})\nabla f(x)\rangle \\
&+ \tfrac{1}{4}\left\|v + 2\sqrt{\mu}(x - x_*) + \sqrt{s}\nabla f(x + tv)\right\|^2 \\
&+ \tfrac{1}{4}\left\|-t\sqrt{s}\nabla^2 f(x)v - t(1 + \sqrt{\mu s})\nabla f(x)\right\|^2 \\
&+ \tfrac{2}{4}\langle v + 2\sqrt{\mu}(x - x_*) + \sqrt{s}\nabla f(x + tv), -t\nabla^2 f(x)v - t(1 + \sqrt{\mu s})\nabla f(x)\rangle \\
&- \tfrac{1}{4}\|v\|^2 - \tfrac{1}{4}\left\|v + 2\sqrt{\mu}(x - x_*) + \sqrt{s}\nabla f(x)\right\|^2.
\end{aligned}
$$

Canceling out the corresponding terms we get

$$
(1 + \sqrt{\mu s})(f(x + tv) - f(x)) + \tfrac{1}{4}\left\|-t2\sqrt{\mu}v - t\sqrt{s}\nabla^2 f(x)v - t(1 + \sqrt{\mu s})\nabla f(x)\right\|^2
$$
$$
+ \tfrac{2}{4}\langle v, -t2\sqrt{\mu}v - t\sqrt{s}\nabla^2 f(x)v - t(1 + \sqrt{\mu s})\nabla f(x)\rangle + \tfrac{1}{4}\left\|v + 2\sqrt{\mu}(x - x_*) + \sqrt{s}\nabla f(x + tv)\right\|^2
$$
$$
- \tfrac{1}{4}\left\|v + 2\sqrt{\mu}(x - x_*) + \sqrt{s}\nabla f(x)\right\|^2 + \tfrac{1}{4}\left\|-t\nabla f(x)v - t(1 + \sqrt{\mu s})\nabla f(x)\right\|^2
$$
$$
+ \tfrac{2}{4}\langle v + \sqrt{s}\nabla f(x + tv), -t\sqrt{s}\nabla^2 f(x)v - t(1 + \sqrt{\mu s}\nabla f(x))\rangle
$$
$$
+ \tfrac{2}{4}\langle 2\sqrt{\mu}(x - x_*), -t\sqrt{s}\nabla^2 f(x)v - t(s + \sqrt{\mu s})\nabla f(x)\rangle.
$$

We focus now on the fourth and the fifth addends which are the most problematic. Using $\|a + b\|^2 = \|a\|^2 + 2\langle a, b\rangle + \|b\|^2$

$$
\underbrace{\tfrac{1}{4}\left\|v + 2\sqrt{\mu}(x - x_*)\right\|}_{1} + \underbrace{\tfrac{1}{4}s\left\|\nabla f(x + tv)\right\|^2}_{2} + \underbrace{\tfrac{2}{4}\langle v + 2\sqrt{\mu}(x - x_*), \sqrt{s}\nabla f(x + tv)\rangle}_{3}
$$
$$
\underbrace{- \tfrac{1}{4}\left\|v + 2\sqrt{\mu}(x - x_*)\right\|^2}_{1} \underbrace{- \tfrac{1}{4}s\left\|\nabla f(x)\right\|^2}_{2} \underbrace{- \tfrac{2}{4}\langle v + 2\sqrt{\mu}(x - x_*), \sqrt{s}\nabla f(x)\rangle}_{3}.
$$

We cancel out the terms with 1 and we bound the terms 3 by

$$
\tfrac{2}{4}\langle v + 2\sqrt{\mu}(x - x_*), \sqrt{s}(\nabla f(x + tv) - \nabla f(x))\rangle \leq \sqrt{s}\tfrac{2}{4}\|v\|^2 Lt + \frac{\sqrt{\mu s}}{\mu}tL\|\nabla f(x)\|\,\|v\|.
$$

For the terms with 2 we get

$$
\begin{aligned}
&= \ \|\nabla f(x + tv)\|^2 - \|\nabla f(x)\|^2 + \langle\nabla f(x + tv), \nabla f(x)\rangle - \langle\nabla f(x + tv), \nabla f(x)\rangle \\
&= \ \langle\nabla f(x + tv), \nabla f(x + tv) - \nabla f(x)\rangle + \langle\nabla f(x + tv) - \nabla f(x), \nabla f(x)\rangle \\
&\leq \ \langle\nabla f(x + tv) - \nabla f(x) + \nabla f(x), \nabla f(x + tv) - \nabla f(x)\rangle \\
&\quad + \langle\nabla f(x + tv) - \nabla f(x), \nabla f(x)\rangle \\
&\leq \ \|\nabla f(x + tv) - \nabla f(x)\|^2 + \langle\nabla f(x), \nabla f(x + tv) - \nabla f(x)\rangle \\
&\quad + \langle\nabla f(x + tv) - \nabla f(x), \nabla f(x)\rangle \\
&= \ \|\nabla f(x + tv) - \nabla f(x)\|^2 + 2\langle\nabla f(x), \nabla f(x + tv) - \nabla f(x)\rangle \\
&\leq \ L^2 t^2 \|v\|^2 + 2\langle\nabla f(x), \nabla f(x + tv) - \nabla f(x)\rangle \\
&\leq \ L^2 t^2 \|v\|^2 + 2\|\nabla f(x)\| Lt \|v\|.
\end{aligned}
$$

Finally, we observe that using

$$
f(x + tv) - f(x) \leq \langle\nabla f(x), tv\rangle + \tfrac{L}{2}\|tv\|^2,
$$
$$
\|x - x_*\| \leq \frac{\|\nabla f(x)\|}{\mu},
$$

in combination with Young's inequality, the term II can be bounded by a function of the form $B_{ST}(p)t + C_{ST}(p)t^2$. That is, we collect the coefficients of the linear terms in $t$ in $B_{ST}(p)$ and the coefficients of the quadratic terms in $t$ in $C_{ST}(p)$. These functions can be chosen to be of the form $\gamma_1 \|v\|^2 + \gamma_2 \|\nabla f(x)\|^2$ for $\gamma_i$ positive real numbers.

• **Term $III$.** We start with

$$F(p) = \frac{\sqrt{s}}{2}(\|\nabla f(x)\|^2 + v^T \nabla^2 f(x)v),$$

and

$F(p + tX_{\mathrm{NA}}(p)) = \frac{\sqrt{s}}{2}(\|\nabla f(x+tv)\|^2$

$+(v - t2\sqrt{\mu}v - t\sqrt{s}\nabla^2 f(x)v - t(1+\sqrt{\mu s})\nabla f(x))^T \nabla^2 f(x+tv)(v - t2\sqrt{\mu}v - t\sqrt{s}\nabla^2 f(x)v$

$-t(1+\sqrt{\mu s})\nabla f(x)))$.

Therefore,

$F(p + tX_{\mathrm{NA}}(p)) - F(p) = \frac{\sqrt{s}}{2}(\|\nabla f(x+tv)\|^2 - \|\nabla f(x)\|^2) + \frac{\sqrt{s}}{2}(v^T(\nabla^2 f(x+tv) - \nabla^2 f(x))v)$

$+\frac{\sqrt{s}}{2}(2v^T \nabla^2 f(x+tv)(-t2\sqrt{\mu}v - t\sqrt{s}\nabla^2 f(x)v - t(1+\sqrt{\mu s})\nabla f(x))$

$+(-t2\sqrt{\mu}v - t\sqrt{s}\nabla^2 f(x)v - t(1+\sqrt{\mu s})\nabla f(x))^T \nabla^2 f(x+tv)(-t2\sqrt{\mu}v$

$-t\sqrt{s}\nabla^2 f(x)v - t(1+\sqrt{\mu s})\nabla f(x)))$.

Notice that a bound for the first addend was computed in term II. The second addend can be bounded using the Lipschitz continuity of the Hessian by $\frac{\sqrt{s}}{2}Ht\|v\|^3$. The remaining terms are easily bounded using Cauchy-Schwartz and Young's inequality. Observe that with the exception of $\frac{\sqrt{s}}{2}Ht\|v\|^3$, we obtain terms that are linear and quadratic in $t$ and their coefficients are linear combinations of $\|\nabla f(x)\|^2$ and $\|v\|^2$. These terms have no correspondence in the heavy-ball case, but the coefficients of the terms linear in $t$ may be added to the function $A_{ST}$ and the coefficients of the terms quadratic in $t$ may be added to $C_{ST}$, as they play an analogous role.

• **Term $IV$.** From the Appendix in [1] one has

$$\begin{aligned}\langle \nabla V(p), X_{\mathrm{NA}}(p)\rangle \quad &\le -\sqrt{\mu}(\|v\|^2 + (1+\sqrt{\mu s})\langle x - x_*, \nabla f(x)\rangle \\ &\quad -\frac{1}{2}\sqrt{s}(v\nabla^2 f(x)v + \|\nabla f(x)\|^2)).\end{aligned}$$

On the other hand

$\alpha V(p) \quad = \alpha((1+\sqrt{\mu s})(f(x) - f(x_*)) + \frac{1}{4}\|v\|^2 + \frac{1}{4}\|v + 2\sqrt{\mu}(x - x_*) + \sqrt{s}\nabla f(x)\|^2)$

$\le \alpha((1+\sqrt{\mu s})(f(x) - f(x_*)) + \frac{1}{4}\|v\|^2 + \frac{3}{4}(\|v\|^2 + \|2\sqrt{\mu}(x - x_*)\|^2 + \|\sqrt{s}\nabla f(x)\|^2)),$

and

$$F(p) = \frac{\sqrt{s}}{2}(\|\nabla f(x)\|^2 + v^T \nabla^2 f(x)v).$$

Following computations analogous to the heavy-ball case, we can easily upper bound $\langle \nabla V(p), X_{\mathrm{NA}}(p)\rangle + \alpha V(p) + F(p)$ by an expression of the form $\gamma_1(\alpha)\|v\|^2 + \gamma_2(\alpha)\|\nabla f(x)\|^2$, where $\gamma_i(\alpha)$ are shown to be negative if $\alpha \le \frac{\sqrt{\mu}}{4}$. These computations allow us to define $D_{ST}$.

**Algorithm and Properties**

Gathering together all the computed bounds of the terms I, II, III, and IV we can go back to (5) and

$$\frac{d}{dt}V(p(t)) + \alpha V(p(t)) + F(p(t)) \le C_{ST}(\hat{p})t^2 + (A_{ST}(\hat{p}) + B_{ST}(\hat{p}))t + D_{ST}(\hat{p}).$$

Observing that for any point $p = [x, v]^T$ such that $\|v\|$ and $\|\nabla f(x)\|$ are not zero, $C_{ST}(p) > 0$ and $D_{ST}(p) < 0$ if $\alpha \le \frac{\sqrt{\mu}}{4}$, it is easy to see that $C_{ST}(p)t^2 + (A_{ST}(p) + B_{ST}(p))t + D_{ST}(p) = 0$ has always a positive solution, which is

$$\text{step}_{ST}(p) = \frac{-(A_{ST}(p) + B_{ST}(p)) + \sqrt{(A_{ST}(p) + B_{ST}(p))^2 - 4C_{ST}(p)D_{ST}(p)}}{2C_{ST}(p)}.$$

---

**Algorithm 1:** Triggered Forward-Euler algorithm

---

**Initialization:** Initial point ($p_0$), convergence rate ($\alpha$), objective function ($f$), tolerance ($\epsilon$);
**Set:** $k = 0$;
**while** $\|\nabla f(x)\| \geq \epsilon$ **do**
  Compute stepsize $t_k = \text{step}_{ST}(p_k)$;
  Compute next iterate $p_{k+1} = p_k + t_k X_{\mathrm{NA}}(p_k)$;
  Set $k = k + 1$
**end**

---

We pointed out how to obtain bounds to design a self-triggered implementation, but its is very easy to obtain an event-triggered implementation from there. Finally, we have the following result.

**Theorem 3.** *For $0 < \alpha \leq \sqrt{\mu}/4$, Algorithm 1 is a variable stepsize integrator with the following properties*

*(i) the stepsize is uniformly lower bounded by a positive, explicit constant;*

*(ii) $\frac{d}{dt} V(p_k + t X_{\mathrm{NA}}(p_k)) \leq -\alpha V(p_k + t X_{\mathrm{NA}}(p_k)) - F(p_k + t X_{\mathrm{NA}}(p_k))$ for all $t \in [0, t_k]$ and all $k \in \{0\} \cup \mathbb{N}$.*

*As a consequence, it follows that $f(x_{k+1}) - f(x_*) \leq \frac{2\|x(0) - x_*\|^2}{s} e^{-\alpha \sum_{i=0}^k t_i}$ for all $k \in \{0\} \cup \mathbb{N}$.*

*Proof.* Due to the similarities with the heavy-ball case we only outline the proof. We have already shown how to compute a positive stepsize at any point, and are left to show that this stepsize is positively lower bounded. In the end, the proof of Theorem 3.3 in the paper "Convergence-Rate-Matching Discretization of Accelerated Optimization Flows Through Opportunistic State-Triggered Control" relies on using twice the bounds provided by expressions of the form

$$\frac{\beta_1 \|v\|^2 + \beta_2 \|\nabla f(x)\|^2}{\beta_3 \|v\|^2 + \beta_4 \|\nabla f(x)\|^2}, \tag{6}$$

which are known to be upper and lower bounded by positive constants when the $\beta_i$ are positive.

Firstly, in the proof of Theorem 3.3, the form (6) is used to lower bound $\frac{-D_{ST}}{C_{ST}}$. This can be done the same way, just with different coefficients $\beta_i$, due to the computations sketched above.

The form (6) is needed a second time to upper bound $\frac{A_{ST}(p) + B_{ST}(p)}{2C_{ST}(p)}$. The only difference is that now $A_{ST}(p)$ includes terms of the form

$$\gamma_1 \|v\|^3, \ \gamma_2 \|v\|^2 \|\nabla f(x)\|. \tag{7}$$

for $\gamma_i \in \mathbb{R}$ which are not linear combinations of the terms $\|v\|^2$ and $\|\nabla f(x)\|^2$ and therefore $\frac{A_{ST} + B_{ST}}{2C_{ST}}$ does not have the form (6). However, this can be easily fixed if we observe that, by design, the iterates produced by Algorithm 1 stay inside the starting sub-level set of the Lyapunov function (3). More precisely, since by construction the value of the Lyapunov function can only decrease along the discrete dynamics, if $p_k = [x_k, v_k]^T$ then

$$\frac{1}{4} \|v_k\|^2 \leq (1 + \sqrt{\mu s})(f(x_k) - f(x_*)) + \frac{1}{4} \|v_k\|^2 + \frac{1}{4} \|v_k + 2\sqrt{\mu}(x_k - x_*) + \sqrt{s}\nabla f(x_k)\|^2$$
$$= V(p_k) \leq V(p_0).$$

Using this we can upper bound the terms in (7) by linear combinations of $\|v\|^2$ and $\|\nabla f(x)\|^2$, for instance $\|v\|^3 \leq \sqrt{4V(p_0)} \|v\|^2$. Thus, we can upper bound $\frac{A_{ST} + B_{ST}}{2C_{ST}}$ by an expression of the form (6) and we can apply Lemma 1. Finally, an argument analogous to the one employed in the heavy-ball case gives the proof. $\square$

# 3  Illustrations on Quadratic Objective Functions

We compare the performance of Algorithm 1 with the explicit and symplectic integrators proposed in [2] in the case of 2-dimensional quadratic objective functions. These examples allow to see the evolution of the state-variables in a neat way. Figure 1 illustrates how the state evolves under the corresponding discrete dynamics for the objective functions $f_1(x_1, x_2) = x_1^2 + x_2^2$ and $f_2(x_1, x_2) = x_1^2 + 10^4 x_2^2$. In both cases, $\alpha = \sqrt{\mu}/4$ and $s = \mu/(16L^2)$ and $s = \mu/(36L^2)$ (according to the values used in [2])

Figure 1: The left plot displays three discrete algorithms converging from $(50, 50)$ to the minimizer $0$ of the function $f_1(x_1, x_2) = x_1^2 + x_2^2$. The stepsize of the ST integrator is significantly larger than the one of the other two methods, which makes it converge in a far fewer number of iterations. We also compare the difference in performance due to different values of $s$, showing that $s = \mu/(16L^2)$ performs slightly better. On the right plot, we run the three algorithms for the same number of iterations (10000) starting at $(50, 5000)$ for the function $f_2(x_1, x_2) = x_1^2 + 10^4 x_2^2$. Due to the large stepsize of the ST integrator the algorithm makes considerable larger progress toward the optimum as compared to the other two methods. The ST integrator corresponding to the value $s = \mu/(16L^2)$ slightly outperforms the one corresponding to the value $s = \mu/(36L^2)$, although it is not noticiable in the plot.

Next, we introduce the optimal stepsize only for comparison purposes, as the minimizer is in practice unknown. Figure 2 compares the stepsizes obtained with a priori knowledge of the location of the optimizer against the ones obtained by the ET- and the ST-integrators, as well as with the constant stepsize for the explicit method in [2].

Figure 2: Evolution of the stepsizes obtained with a priori knowledge of the optimizer, the ones obtained by the ET- and the ST-integrators, and the constant stepsize for the explicit method in [2]. Left, from $(50, 50)$ and right, from $(500, 50)$. The objective function is $f(x_1, x_2) = 10x_1^2 + 10^3 x_2^2$.