[Reviews · NeurIPS 2019]

Reviewer 1



This paper studies how to discretize continuous-time ODEs into discrete-time accelerated optimization methods in a rate-maintaining manner. The authors propose a solution by borrowing the concept of opportunistic state-triggered control from the control. The proposed approach makes use of the Lyapunov functions for the so-called high-resolution differential equations to design variable-stepsize forward-Euler discretizations. Originality: The idea of this paper is definitely original. Although the paper uses some results in a recent paper (Ref[22]) which introduces the high-resolution ODE, the connection between state-triggered control and optimization design seems new and novel to me. This is the part I really like about the paper. Quality: The technical quality of this paper are reasonably good (especially on the theory side). I haven't noticed any technical flaws. However, I do have the following concerns. 1) The rate obtained by the proposed approach seems very weird to me. I looked at the results in Theorem 3.3. How does this result even recover the standard rate (1-sqrt(mu/L)) where L is the smoothness constant? At least in the first glance the bound in Theorem 3.3 does not even depend on L. I can not see how eventually a factor of sqrt(1/L) will come out from the bounds in Theorem 3.3. As a sanity check, at least Theorem 3.3 should recover the standard discrete-time accelerated rate 1-sqrt(mu/L). In continuous time, the constant L won't show up and that is as expected. Once the discretization is involved, the constant L will play a role. I can see the terms in Proposition 3.2 depend on L but just cannot see how the final convergence bound in Theorem 3.3 depends on L at this point. 2) The numerical examples need some improvements. A 2-dimensional quadratic example is not going to be that convincing. In addition, both Figures 2 and 3 are confusing. The right plot in Figure 2 seems some divergence behavior, right? I am very confused here. Figure 3 plots the case where the optimal point is already known. I don't think this is a good plot demonstrating the potential of the proposed approach. Please explain what exactly happens in Figure 2. Using the same amount of iterations, does the proposed method outperform existing discretization methods? If so, making a plot to precisely show that. Clarity: Overall the paper is well written. As I mentioned before, Figure 2 confuses me a lot. Other than that, I think the paper is quite clear. Significance: Bridging state-triggered control with optimization design is a novel idea that may inspire further research. I think the conceptual novelty of this paper is high, although I am a little bit unsure whether the proposed discretization approach itself really outperforms the existing approaches. A minor comment: The authors claim that their approach is general for any dynamical systems. Is this true? It seems the construction of the g function will be case-dependent and hence applying the proposed approach to general dynamical systems seems to require significant amount of extra work. ==================================================== After the rebuttal: The analytical comparison between the proposed discretization and existing approaches is still not made at this moment. It will be much more convincing if the proposed approach can at least recover the standard rate of Nesterov's method. I also agree with Reviewer 2 that there is a concern on whether preserving the continuous time rate itself is the most important thing to do or not. Overall this paper has something I really like (the original idea of bridging optimization and state/event-triggered control) but at the same time the true benefits of the proposed approach are not convincing. I tend to keep my score as weak acceptance.

Reviewer 2



Please, see comments in item 1. A couple of other comments are below. The numerical implementation of the proposed method requires computing all the quantities stated in Proposition 3.2 (in my opinion this is quite cumbersome). Moreover, one might converge in fewer iterations but it is also necessary to compute more gradients and several other quantities. Thus, the number of iterations is not totally representative of the computational complexity. The numerical experiments illustrate a trivial case, quite far from real problems encountered in machine learning or statistics. Finally, this paper does not make due justice to several other recent papers on the topic, which are not even mentioned. For instance, I suggest the authors check recent papers from: Jordan, Attouch, Franca, Vidal, Cabot, etc. (to cite just a few). ============== post authors response: "We respectfully disagree with this comment, as we include the work by 9 the authors suggested by R#2, see [3], [22], [23], [28] and [29]." Obviously, I wasn't referring to these papers which are already cited. An exhaustive literature review is not what I mean. I just wanted to point out that some consideration to existing literature may be appropriate. Replacing the Hessian by the mentioned approximation, $\nabla f(x_{k+1}) - \nabla f(x_k)$, introduces another source of error in the discretization. I cannot judge whether this is meaningful in this case without seeing the proper derivation. I agree that the idea of the paper is interesting, and deserves explorations. However, in my opinion the current results of the paper are insufficient. Moreover, I don't see any benefits compared to much simpler, elegant, and so far more efficient approaches. Let me stress another point of concern. There are many methods in ODEs literature to implement discretization designed to preserve some given quantity, such as the energy of the systems for instance. One can simply use any discretization and project the solution. It turns out that such approaches are much inferior since they are considerably more unstable than simpler methods which do not preserve the same quantity. Thus, I'm not so sure that just because one can force the discretization to preserve the decaying of the Lyapunov function, this automatically will lead to a more efficient method. This has to be justified through stability analysis.

Reviewer 3



==================post rebuttal=========================== I'd like to thank the authors' for their detailed responses. However, it seems that they didn't completely understand the questions related to the benefit/importance of opportunistic state-triggered control and the limitation of the numerical experiments. The benefit should be compared to other existing approaches like Nesterov's acceleration and heavy-ball methods, and should be analytical and quantitative, instead of just comparing with fixed step-sizes and simply high-level intuition. And for the numerical experiments, the issue is not just about sizes (like increasing from dimension 2 to dimension 50), but about the generality of the function classes (e.g., at least the authors should consider regularized logistic regression, etc.). These being said, I still like the idea of the paper, and the way they derive the state-triggered step-sizes. However, the limitation in comparisons with other existing approaches and the practical applicability of this approach limits its significance, and as a result I would like to maintain my score. ======================================================= This paper focuses on the ODE viewpoint of the acceleration methods in optimization, and introduces the idea of opportunistic state-triggered control to design efficient discretization of the ODE systems, aiming at preserving the acceleration properties of the (high resolution) ODE systems. Resorting to the Lyapunov function analysis of the corresponding ODE systems and convexity arguments, the authors propose a practical and simple variable-stepsize forward-Euler discretization to the high-resolution ODEs obtained for heavy-ball and Nesterov algorithms recently, yielding an efficient algorithm with convergence rate matching those of the corresponding ODEs. Here the step sizes are decided utilizing the Lyapunov function, as is done in the opportunistic state-triggered control. Finally, some very simple numerical experiments are provided to validate the efficiency of the proposed algorithm. The originality of the paper is basically beyond doubt, especially in the sense of connecting the opportunistic state-triggered control with acceleration methods in optimization. However, the level of significance of the paper is relatively limited due to the following issues: 1. The authors do not compare the bound obtained in Theorem 3.3 with the existing ones, e.g., those obtained in [23]. What do we actually benefit from the opportunistic state-triggered control, or variable-step size discretization in theory? In addition, do we have to restrict the application of such discretization approaches to high resolution ODEs, as is the case of symplectic discretization as found in [23]? What would happen if we apply the opportunistic state-triggered control discretization to the low-resolution ODEs, e.g., those in earlier papers like [24]? Without answering these questions, the theoretical significance of the new discretization/algorithm is a bit limited. 2. The numerical experiments are too limited. The authors may want to consider some more interesting numerical examples, to showcase the improvement of the new algorithm compared to the existing ones. The quality of the paper is also generally good, despite the issues mentioned above, which kind of limit the completeness the theory and the applicability in practice related to the proposed approach. The clarity and writing of the paper is pretty good, making it generally an enjoyable reading experience. Some minor suggestions include: 1. In Section 2.2, when defining V, the authors may want to clearly write it as V(X(x,u)), and explain that the derivative in \dot{V} is w.r.t. u, which leads to the first equality in (2). The authors may also want to explain more about what the control u=k(x) serves for, especially given that u disappears in the subsequent sections starting from Section 2.3. Otherwise, this part seems to be a little bit confusing even to a reader with some background in all the related areas. T Finally, some typos and minor suggestions: 1. Line 69: “[28]” -> “[28], “ — a comma is missing. 2. Line 106: “not” seems to be “now”? 3. Line 117: “criterium” might better be “criterion”. 4. Lines 124-125: the authors may want to mention that g is continuous, since otherwise the definition of t_{i+1} may not ensure no zero-crossing between (t_i, t_{i+1}). 5. Lines 123-130: the terminology “state-triggered”, “event-triggered” (ET) and “self-triggered” (ST) seems to be a little bit confusing, especially given the latter parallel presentation of ET and ST in the algorithm section. However, according to the description of lines 123-130, it seems that ST is a special case of ET, instead of two parallel concepts? And is “event-triggered design” a special case of state-triggered control? If so, what do “event” and “state” refer to, respectively? The authors may want to explain these more intuitively and clearly, so as to avoid potential confusion. 6. Line 189: “with a slight abuse of notation” — this sentence is a bit confusing. If it means that in Section 2.2 g is a function of (x_1,x_2) (same dimension for two arguments), while here the second argument becomes (p,t) (first argument multi-dimensional, and second argument single-dimensional), then this could be stated and discussed earlier in Section 2.2, so that readers understand that g can be more flexible. 7. Lines 190-191: “becomes apparent below” — does it refer to Lines 194-195? If so, then just state it immediate after the claim, not “below”. 8. Numerical experiments: why is ET integrator missing in Figure 2? And what is the observation and conclusion for Figure 3 (the authors seem to only state what is plotted in Figure 3 as it is, without drawing any concrete conclusions or comments)?

Reviewer 4



Summary: ------------- The paper applies stage-triggered control to discretize ODEs of optimization methods. The approach is applied for the Heavy-ball method where a rate is presented together with empirical illustrations. Strengths: -------------- -The paper is very original, the techniques used are new and relevant for optimization. - The event triggered step-size is potentially adaptive to values of the function in a principled way. This leads to interesting application of the ODEs interpretations. Though the current work is limited to the strongly convex case, the approach is sufficiently original to appeal generalizations from the community. The real challenges to me in optimization are adaptivity or generalizations to the non-convex case. This paper may provides a principled way for those challenges. Weaknesses: ----------------- - The paper is hard to parse. - There is a real problem in the bound: dependence of s for the lower bound in the step-size is not explicit. One really needs it to get an actual principled implementation of the algorithm and an understanding of their bound. For the moment it is unclear if the rate is accelerated or not. I think the authors should withdraw the claim that they "preserve the acceleration properties of high-resolution differential equations" as long as they do not provide an explicit lower bound on the step-sizes. - The results for Nesterov acceleration are unclear: there is no need of assumptions in the Hessian a priori. Conclusion: ---------------- I would accept the paper for the novelty of the approach and the results. This is an interesting direction of work that does not simply attempt to reproduce previous results but explores a new procedure to get step-sizes of accelerated methods. This may lead nowhere but the present results are sufficient for me to open the discussion in a conference. After discussion -------------------- I read the author's response and the discussion. Though I agree that the results do not yet match optimal results in the litterature, I think that the idea can open other directions (application for stochastic optimization for example). This paper would not be accepted for a journal obviously but its originality is what a conference like Neurips is seeking in my opinion. Therefore I maintain my score. Yet, if accepted, the authors need to provide an explicit rate. The present work is not complete without it and it would greatly diminish its impact.

[Author Response · NeurIPS 2019]

**Relationship between the constant $L$ and convergence rate, comparison w/ [23] (Reviewers $\#1$ and $\#3$).** Th 3.3 (line 202) relates the algorithm convergence rate to the stepsize $t_k$. This stepsize depends on the constant $L$ through the expression of $\text{step}_\#(p)$, which shows the dependency between convergence rate and $L$. In the paper, we prove that $\text{step}_\#(p)$ is lower bounded. We have observed numerically that the stepsize is larger than the stepsizes used by other discretizations schemes in the heavy-ball method, as shown in Fig 3, and will provide further numerical evidence in the revised version. We are currently working on an analytical comparison with [23], which requires the explicit computation of a tight lower bound of $\text{step}_\#(p)$ as a nonlinear function of $s$, $\alpha$, $\mu$, and $L$ (R$\#3$).

**Omitted relevant literature (Reviewer $\#2$).** We respectfully disagree with this comment, as we include the work by the authors suggested by R$\#2$, see [3], [22], [23], [28] and [29]. It is impossible to provide an exhaustive literature review, but we will include recent papers[1] (available after the submission of our work) by Attouch, França, and co-authors which deal, resp., with the discretization of inertial systems with Hessian-driven damping and conformal Hamiltonian systems to obtain optimization algorithms. We will include Kolarijani *et al.*, which uses hybrid dynamical systems to generate fast optimization methods that employ constant-stepsize discrete dynamics. The differences with our work are clear, as none of these references design variable-stepsize integrators based on event-triggered control.

**Perceived limited applicability of the proposed setting and extensions beyond it (all reviewers).** As a result of the concerns raised by R$\#2$, we have realized that the twice differentiablity assumption can be weakened: in the heavy-ball case, only continuous differentiablity is needed for the discretization. In Nesterov's case, twice differentiability arises from the presence of a Hessian term $\sqrt{s}\nabla^2 f(x)v$ in the ODE, which is inherited by the discretization. The work [23] replaces it by $\nabla f(x_{k+1}) - \nabla f(x_k)$ when discretized, providing an appealing research direction circumventing the use of the Hessian. It is standard practice in the literature to assume knowledge of $\mu$ and $L$ for strongly-convex functions when looking for the optimal rate. Besides, several methods have been designed to approximate these constants in practice, and they can surely be adapted to our setting. As pointed by R$\#1$, the function $g$ is case-dependent, but the methodology presented here is applicable to the discretization of other dynamical systems endowed with a Lyapunov function certificate. We agree with R$\#3$ that pursuing this will broaden the applicability of our theory. Although regularization can also be used to endow convex functions with strong convexity, it would also be extremely interesting to extend this methodology to the convex framework (R$\#2$). Nonetheless, the main point of the paper is to introduce the idea of a systematic way to develop discretizations that maintain the convergence rate properties of their continuous counterparts. R$\#1$ and $\#3$ point out that the originality of the paper is " basically beyond doubt" and we believe it may inspire new research given the recent explosion of activity in the area of high-resolution ODEs.

**Importance of opportunistic state-triggered control and variable-stepsize discretization (Reviewer $\#3$).** Opportunistic state-triggered control saves resources by taking into account the current system state while maintaining performance guarantees. This is in contrast to periodic sampling, where worst-case scenarios have to be taken into account, drastically reducing inter-sampling time. Analogously, the proposed integrators take into account the current state of the dynamics through the values of $v$ and $\nabla f$ to adjust its stepsize while satisfying convergence and performance guarantees. This contrasts with fixed-stepsize integrators, whose stepsize is limited by the most unfavorable situation. In practice, this may have a critical impact on performance. We will address any possible confusion (especially regarding terminology) pointed by R$\#3$ in the revised version.

**Simulations (all reviewers).** We will include richer numerical experiments if the paper is accepted. We have run now simulations with quadratic functions defined by 50x50 matrices with similar results. *Convergence* will be shown by plotting the decay of the objective and Lyapunov functions (R$\#1$). Regarding *Fig* 2, we show that the three discretization procedures follow the same trajectory (the continuous dynamics). The proposed approach is able to follow the curve taking longer stepsizes, thus making further progress when run for an equal number of iterations. Formally, let us denote by $t_k$ the stepsize of our method at iteration $k$ and by $s$ the stepsize of a fixed-stepsize integrator. After $n$ iterations, our integrator approximates the continuous dynamics at $\sum_{k=1}^{n} t_k$, while the constant-stepsize integrators approximates it at $n \cdot s$. In simulations, $\sum_{k=1}^{n} t_k$ is significantly larger than $n \cdot s$ (R$\#1$). We will also include the ET integrator for comparison in Fig 2 in the revised version (R$\#3$). Finally, we introduce the *optimal stepsize* only for comparison purposes, as the minimizer is in practice unknown. Knowledge of the minimizer $x_*$ would enable the explicit computation of the Lyapunov function (cf. Th. 3.1), which in turn allows to solve $\dot{V} + \alpha V = 0$ (cf. line 181) by any standard numerical method at any iteration. We refer to this solution as optimal stepsize (Fig 3, green), as is the actual largest stepsize one may take conserving the Lyapunov decay. Fig 3 illustrates how our algorithm is able to chase this optimal stepsize at any iteration, without knowledge of the minimizer (R$\#1$ and $\#3$). R$\#2$ also points out that the computation of the stepsize may be convoluted. While the ET integrator is more involved, the ST integrator relies on a simple function of the quantities $\|v\|$, $\|\nabla f\|$ and $\langle v, \nabla f \rangle$, (see $\text{step}_{ST}$, line 196) which can be computed easily.

## Footnotes

[1]H. Attouch, Z. Chbani, J. Fadili, and H. Riahi. First-order optimization algorithms via inertial systems with Hessian driven damping. *arXiv:1907.10536*, 2019; G. França, J. Sulam, D. Robinson, and R. Vidal. Conformal symplectic and relativistic optimization. *arXiv:1903.04100*, 2019; A. S. Kolarijani, P. M. Esfahani, and T. Keviczky. Fast Gradient-Based Methods with Exponential Rate: A Hybrid Control Framework. In *Proceedings of the 35th International Conference on Machine Learning*, 2018


[Meta-Review · NeurIPS 2019]

This paper proposes to use ideas from opportunistic state-triggered control to discretize ODEs for accelerated optimization algorithms. Several reviewers praised the originality of the work, stating that "the originality of the paper is basically beyond doubt" and that "the techniques used are new and relevant for optimization". Some reviewers also expressed concerns about the comparison of the results to ones from previous works, for instance comparing "the bound obtained in Theorem 3.3 with the existing ones, e.g., those obtained in [23]". The concerns also included how stringent the assumptions were, in particular for the results for the Nesterov case and the need for additional experimental results. The authors submitted a detailed response to the reviewers' comments. After reading the response and updating their reviews, the reviewers feel that the paper would greatly benefit from a detailed quantitative discussion and comparison of the obtained bounds with those from [23]. The reviewers also feel that the twice differentiability assumption should not be needed for the Nesterov case; this deserves further inspection in order to assume continuous differentiability instead. An additional expert opinion was sought, leaning towards "accepting the paper for the novelty of the approach and the results". All in all, the originality of this work, within this active area of research, is clear. Furthermore, while the current results may seem improvable, the proposed viewpoint and techniques open new venues for research in the area of high-resolution ODEs for first-order optimization. We strongly recommend to the authors to take the reviewers' comments and suggestions into account while preparing the camera-ready final version of the paper. Accept.